# Learnability of Discrete Dynamical Systems under High Classification Noise

## Abstract

Discrete dynamical systems are principled models for real-world cascading phenomena on networks, and problems for learning dynamical systems have garnered considerable attention in ML. Existing studies on this topic typically assume that the training data is *noise-free*, an assumption that is often impractical. In this work, we address this gap by investigating a more realistic and challenging setting: learning discrete dynamical systems from data contaminated with *noise*. We present *efficient* noise-tolerant learning algorithms that provide provable performance guarantees, and establish tight bounds on sample complexity. We show that, even in the presence of noise, the proposed learner only incurs a marginal increases in training set size to infer a system. Notably, the number of training samples required in the noisy setting is the *same* (to within a constant factor) as the upper bound in the noise-free scenario. Further, the number of noisy training samples used by the algorithm is *only a logarithmic factor* higher than the best-known lower bound. Through experimental studies, we evaluate the empirical performance of the algorithms on both synthetic and real-world networks.

## 1 Introduction

**Background and Motivation.** Discrete dynamical systems serve as formal models for various real-world diffusion processes on networks, including the spread of rumors, information, and diseases (Battiston et al., 2020; Ji et al., 2017; Lum et al., 2014; Sneddon et al., 2011; Schelling, 2006; Laubenbacher & Stigler, 2004; Kauffman et al., 2003). For dynamical systems in the real world, however, one often *cannot* readily obtain a full system specification. Thus, learning unknown components of dynamical systems is an active research area (Chen et al., 2021; Chen & Poor, 2022; Conitzer et al., 2022; 2020; Rosenkrantz et al., 2022; He et al., 2016; Narasimhan et al., 2015; Dawkins et al., 2021; Adiga et al., 2019).

In essence, a discrete dynamical system consists of an underlying *network* over which a contagion spreads. Vertices in the network are entities such as individuals, and edges denote their relationships. To model a cascade, each vertex has a *contagion state* and an *interaction function*. As the contagion spreads, the states of vertices evolve in discrete time steps based on the mechanism of interaction functions. Thus, the interaction functions play an important role in the system dynamics as they model the **behavior** of individuals. One concrete example is the *threshold* function, a classic model for social contagions (Granovetter, 1978; Watts, 2002; Li et al., 2020; Trpevski et al., 2010; Rosenkrantz et al., 2022; Chen et al., 2021). Here, each individual adopts a contagion (e.g., believes a rumor) only when the number of its neighbors adopting this contagion reaches a tipping point.

Our work focuses on learning discrete dynamical systems where the *interaction functions are unknown*. Notably, existing methods for learning interaction functions (e.g., (Adiga et al., 2019; Qiu et al., 2024a)) were developed under one critical assumption: the training data is *noise-free*. However, it is widely recognized that real-world data are often contaminated by *noisy labels* (i.e., classification noise) (Sarfraz et al., 2021; Natarajan et al., 2013; Cesa-Bianchi et al., 1999; Gupta & Gupta, 2019; Kearns & Vazirani, 1994; Angluin & Laird, 1988). In particular, the presence of noise in training data can significantly degrade the prediction accuracy of learning algorithms that are not noise-tolerant. Nevertheless, noise-tolerant learning for discrete dynamical systems has *not* received attention in the literature. In this work, we address this gap with a systematic study of **learning discrete dynamical systems under classification noise**.

**Problem description.** There is a target ground-truth dynamical system where all the interaction functions of vertices are **unknown**. A learner must infer the missing functions from only snapshots of the system's dynamics (provided in a training set) which specify the evolution of vertices' states. Moreover, the training set is contaminated by classification noise (Angluin & Laird, 1988) such that in each snapshot of the dynamics, the updated state of each vertex is incorrect with some (unknown) probability. Our goal is then to design noise-tolerant algorithms that learn all the unknown interaction functions and recover a system that models the behavior of the true system, with performance guarantees under the Probably Approximately Correct (PAC) model (Valiant, 1984).

**Challenges.** Empirical risk minimization (ERM) is often a promising method under the PAC framework. However, the results in (Adiga et al., 2017) (Theorem 4 on page 132) show that in our setting, ERM **cannot** be done efficiently unless **P = NP**. In particular, one **cannot** even efficiently approximate the problem of constructing a system that is consistent with the maximum number of training samples. The second difficulty is that, in our noise setting (see Section 2), the probability of a *multi-class label* (a vector of the states of vertices) being *incorrect* in the training set asymptotically **goes to one** as the system size increases. In other words, *almost surely all the multi-class labels in the training set are wrong*.

Given the importance of the problem and the challenges, we aim to answer the following questions about learning discrete dynamical systems: 1. *Is efficient learning possible under classification noise*? 2. *How many additional samples do we need compared to the noise-free case*?

**Our contributions.** Despite the challenges, we answer both questions. In particular, we show that one can still *efficiently* learn discrete dynamical systems under high classification noise. Notably, the number of training samples required is the **same** (to within a constant factor) as the upper bound in the *noise-free* scenario, and it is only a **log factor higher** than the best-known lower bound.

- **Efficient learnability.** Formally, we propose two *efficient* noise-tolerant algorithms V-ERM and VisRange with respective theoretical and empirical advantages. Both algorithms achieve the PAC guarantee: w.p. at least $1 - \delta$, the prediction error is at most $\epsilon$, for any $\epsilon, \delta > 0$. However, Algorithm V-ERM uses $O(n^2 \log(n))$ training samples, whereas Algorithm VisRange uses only $O(n \log n)$ samples.
  From a theoretical perspective, VisRange wins: the corresponding bound $O(n \log(n))$ on the sample complexity is close to optimal; it matches (to within a constant factor) the information-theoretic *upper* bound in the *noise-free* scenario (Haussler, 1988; Laird, 2012). That is, the algorithm VisRange is strongly noise-tolerant, with only a marginal increase in the number of samples used in comparison with the noise-free case. Further, its sample complexity is only a factor $O(\log n)$ larger than the general *lower* bound in the noisy setting (Angluin & Laird, 1988).

- **Experimental evaluation.** We conduct an experimental study of the algorithms over both real and synthetic networks. Our results highlight a drastic difference in their empirical behaviors: VisRange exhibits a *phase transition* w.r.t. the error rate when the size of the training set reaches a critical threshold. This phenomenon is expected from our theoretical analysis. In contrast, V-ERM shows a steady increase in learning accuracy as more training samples are given. Overall, our experiments underscore that V-ERM is the preferred empirical algorithm due to its simplicity and consistent performance. We then further explore the property of V-ERM w.r.t. different model parameters such as network size, density, and error rates.

**Related work.** The random classification noise (RCN) model is a classic framework for learning under noise, introduced by Angluin and Laird (1988) where they proved the efficient learnability of CNF formulas with at most $k$ literals per clause. Many other learning algorithms under the RCN model have been developed for different concept classes (see e.g., Decatur (1997); Sakakibara (1991; 1993); Kearns & Schapire (1990); Decatur & Gennaro (1995)). Further, both lower and upper bounds on the sample complexity for learning under classification noise are established in Simon (1993); Aslam & Decatur (1996); Laird (2012); Mukhopadhyay & Banerjee (2020). Several other noise models have also been considered (e.g., Kearns & Li (1993); Kearns (1993); Jabbari et al. (2012); Natarajan et al. (2013); Bshouty et al. (2002); Diakonikolas et al. (2019)).

Rigorous methods have been proposed for learning various components of a dynamical system, such as the interaction functions, edge parameters, infection sources, and contagion states, from system dynamics Lokhov (2016); Conitzer et al. (2020); Chen & Poor (2022); Conitzer et al. (2022); Dawkins et al. (2021); Wilinski & Lokhov (2021); Kalimeris et al. (2018); Wen et al. (2017); He

et al. (2016); Narasimhan et al. (2015); Daneshmand et al. (2014); Du et al. (2014); González-Bailón et al. (2011); Hellerstein & Servedio (2007); Li et al. (2020); He et al. (2020); Santos et al. (2024); Sinha et al. (2023). The problem of learning the underlying system topology has also been examined Huang et al. (2019); Pouget-Abadie & Horel (2015); Abrahao et al. (2013); Du et al. (2012); Myers & Leskovec (2010); Gomez-Rodriguez et al. (2010); Soundarajan & Hopcroft (2010). To the best of our knowledge, the problem of learning discrete dynamical systems under classification noise has *not* been studied in the existing literature.

## 2 PRELIMINARIES

### 2.1 DISCRETE DYNAMICAL SYSTEMS

A *discrete dynamical system* over domain $\mathbb{B} = \{0, 1\}$ is defined as a pair $h^* = (\mathcal{G}, \mathcal{F})$, where $(i)$ $\mathcal{G} = (\mathcal{V}, \mathcal{E})$ is an underlying undirected graph with $n$ vertices (e.g., individuals in a social network); $(ii)$ $\mathcal{F} = \{f_v : v \in \mathcal{V}\}$ is a set of *interaction functions*, where $f_v$ is the function for $v \in \mathcal{V}$.

**Interaction functions**. Each vertex in $\mathcal{G}$ has a *state* from domain $\mathbb{B}$ representing its contagion state (e.g., inactive or active). Starting from the initial states of vertices, a system $h^*$ evolves over *discrete time*, with vertices updating their states *synchronously* using the interaction functions. Specifically, for any interaction function $f_v$, the inputs are the current states of $v$'s neighbors; the output of $f_v$ is the next state of $v$. In this work, we focus on dynamical systems with *threshold* interaction functions. Such systems are fundamental models for the spread of **social contagions** such as rumors and information (Granovetter, 1978; Watts, 2002; Li et al., 2020; Trpevski et al., 2010; Rosenkrantz et al., 2022; Chen et al., 2021).

Formally, each vertex $v \in \mathcal{V}$ has an integer threshold $\tau_v^* \in [0, \deg_v + 1]$, where $\deg_v$ is the degree of $v$ in $\mathcal{G}$. At each time $t \geq 1$, the function $f_v$ computes $v$'s state at the next time-step $t + 1$ as follows: $f_v$ outputs (state) 1 if the number of state-1 vertices in $v$'s neighborhood at time $t$ is at least $\tau_v^*$; $f_v$ outputs (state) 0 otherwise. In the rumor-spreading example, a person's belief changes when the number of neighbors believing in the rumor reaches a certain tipping point. An example of a threshold dynamical system is shown in Figure 4 in Appendix A.2.

**Configurations.** A *configuration* $\mathcal{C}$ of a system $h^*$ is a length-$n$ binary vector specifying the contagion *states* of all vertices; here, $\mathcal{C}[v]$ is the state of $v$ under $\mathcal{C}$. Thus, one can view the evolution of system $h^*$ as a time-ordered trajectory of configurations. In a trajectory, a configuration $\mathcal{C}'$ is the **successor** of $\mathcal{C}$ if the system evolves from $\mathcal{C}$ to $\mathcal{C}'$ in a single time-step, denoted by $\mathcal{C}' = h^*(\mathcal{C})$.

### 2.2 LEARNING UNDER NOISE

Let $h^*$ be a ground-truth discrete dynamical system. The learner is provided with *incomplete* information about the system $h^*$, where the underlying graph is known, but all the interaction functions are **unknown**. In this case, the **hypothesis class** $\mathcal{H}$ consists of all systems with the same graph as $h^*$, over all possible threshold assignments for vertices in $\mathcal{V}$. By observing the *noisy* snapshots of system dynamics (given in a training set), the learner's goal is to infer the missing interaction functions and learn a system $h \in \mathcal{H}$ that closely approximates the behavior of the true $h^*$.

**The noisy training set.** Our algorithms learn from a training set that consists of noisy snapshots of the true system $h^*$'s dynamics, under the PAC framework. In particular, we extend the well-known Random Classification Noise (RCN) model (Angluin & Laird, 1988) for binary classification to our multi-class learning context. Let $\mathcal{N} = \{\eta_v : v \in \mathcal{V}(\mathcal{G})\}$ be a collection of **unknown** noise rates where each $\eta_v$ satisfies $0 < \eta_v \leq \bar{\eta} < 1/2$; here, $\bar{\eta}$ is an upper bound on all noise parameters. A **noisy training set** $\mathcal{T} = \{(\mathcal{C}_j, \hat{\mathcal{C}}_j)\}_{j=1}^q$ is formed as follows. For each data point $(\mathcal{C}_j, \hat{\mathcal{C}}_j) \in \mathcal{T}$,

1. The configuration $\mathcal{C}_j \sim \mathcal{D}$ is sampled independently from an **unknown** distribution $\mathcal{D}$. Let $\mathcal{C}_j'$ be the *true* successor of $\mathcal{C}_j$, produced by the unknown ground truth system $h^*$.

2. The learner does **not** see the true successor $\mathcal{C}_j'$. Instead, the observed successor $\hat{\mathcal{C}}_j$ in $(\mathcal{C}_j, \hat{\mathcal{C}}_j) \in \mathcal{T}$ is a *noisy* version of $\mathcal{C}_j'$ where the value of each entry $\mathcal{C}_j'[v]$ is *altered* with probability $\eta_v$. Formally, for each $v \in \mathcal{V}(\mathcal{G})$: $(i)$ $\hat{\mathcal{C}}_j[v] = \neg \, \mathcal{C}_j'[v]$ w.p. $\eta_v$, and $(ii)$ $\hat{\mathcal{C}}_j[v] = \mathcal{C}_j'[v]$ w.p. $1 - \eta_v$.

   As in the RCN model, errors are introduced here by a random process that is independent of the sampling step over $\mathcal{D}$ (Angluin & Laird, 1988). Let $\mathcal{O}$ denote the noisy oracle described above.

For simplicity, we use $(\mathcal{C}, \hat{\mathcal{C}}) \sim \mathcal{O}$ to denote a training pair generated by the oracle $\mathcal{O}$, such that $\mathcal{C}$ is first sampled from $\mathcal{D}$, then the random noise is applied to $\mathcal{C}'$ to get $\hat{\mathcal{C}}$. We use $\mathcal{T} \sim \mathcal{O}^q$ to denote the sampling of a size-$q$ training set $\mathcal{T}$ from $\mathcal{O}$.

Note that $\eta_v$ is often fixed w.r.t $n$. Thus, we sometimes omit the terms involving $\eta_v$ (or $\bar{\eta}$) in our bounds for clarity, and focus on the expressions with the dominating terms given as a function of $n$.

**Remark 1.** In our learning under noise model, the learner is **not** involved in generating the training set. That is, the learner does not generate training data or sample the data; rather, the data are provided to our model. Therefore, it does **not** have control over how the noise is applied to each vertex, and it also does **not** know which vertex's state is wrong in any training sample. Further, the learner does **not** have the exact value of each noise parameter $\eta_v$, $v \in \mathcal{V}$. It is important to note that our multi-class learning model is different from the standard Random Classification Noise (RCN) model. In particular, in the RCN model, strictly *less than half* of the labels are incorrect in expectation. On the other hand, in our setting, the probability of a multi-class label (i.e., the successor of a configuration) in the training set being *incorrect* (i.e., altered by noise) asymptotically goes to one as $n$ increases. In other words, almost surely all the labels in the training set are wrong.

**Remark 2.** In real-world scenarios, the training set $\mathcal{T}$ can be viewed as a collection of snapshots of the true system dynamics: these snapshots can exhibit correlations (w.r.t. the system dynamics), as one snapshot might be the immediate predecessor of another in a trajectory. Consequently, learning from a trajectory of the system evolution can be cast as a special case of our setting. Specifically, suppose $\mathcal{T}$ consists solely of configurations on a trajectory $P$. Then the underlying sampling distribution (unknown to the learner) is such that only configurations on $P$ are sampled with positive probability, while all other configurations are sampled with probability $P$. In this work, we present learners that work under **arbitrary** sampling distributions, including the one mentioned above.

**Remark 3.** We now discuss the model parameters where assumptions are made:

1. **Noise rate** $\eta_v < 1/2$. As highlighted on Page 6, Section 2.1 of (Angluin & Laird, 1988), under the Random Classification Noise Model, when the noise rate equals $1/2$, the errors in the noise process *destroys* all information about the underlying true hypothesis in the training set. As a result, **nothing** can be learned in a meaningful manner and noise-tolerant learning becomes impossible if $\eta_v = 1/2$. This is also discussed in (Kearns & Vazirani, 1994).

   Next, when $\eta_v > 1/2$, note that the problem is **equivalent** to our $\eta_v < 1/2$ case due to symmetry. In particular, in the $\eta_v > 1/2$ regime, one can simply take the complement of the noisy label for each individual vertex, which leads to an error of $1 - \eta_v < 1/2$. This is also discussed in (Angluin & Laird, 1988). Due to these reasons, in our setting (and also in the existing work on random classification noise model), having $\eta_v < 1/2$ represents the **most general form**.

2. **Graph is known**. As shown in (Qiu et al., 2024a), when the graph is *unknown*, one **cannot** efficiently learn the interaction function of dynamical systems even in the noise-free scenario. It immediately follows that the problem remains intractable in our noisy case. Therefore, if efficient noise-tolerant learning of interaction functions is possible, one needs to assume that the underlying graph structure is known.

**Learning from the noisy training set.** Even though the training set $\mathcal{T}$ is from the noisy oracle $\mathcal{O}$, the aim of the learner *remains to find an appropriate hypothesis w.r.t the **true (unknown) distribution** $\mathcal{D}$. Given a new $\mathcal{C} \sim \mathcal{D}$, a learned hypothesis $h$ should predict the true successor $\mathcal{C}'$ (i.e., $h^*(\mathcal{C})$) with high probability. Formally, we use $err_{\mathcal{D}}(h) = \Pr_{\mathcal{C} \sim \mathcal{D}}[h(\mathcal{C}) \neq h^*(\mathcal{C})]$ to denote the **error** admitted by $h$. Note that in our setting, a prediction is considered incorrect if $h(\mathcal{C})$ and $h^*(\mathcal{C})$ *disagree on the state of at least one vertex*; that is, we want to predict the output states of all vertices correctly.

Following the PAC model, for any parameters $\epsilon, \delta > 0$, a learner should find a hypothesis $h \in \mathcal{H}$ such that with probability at least $1 - \delta$ over $\mathcal{T} \sim \mathcal{O}^q$, the error $err_D(h) \leq \epsilon$. The minimum number of training examples required by any learner to achieve the above PAC-guarantee is known as the **sample complexity** of the class $\mathcal{H}$.

## 3 ELEMENT-WISE ERM

We prove the efficient learnability of threshold discrete dynamical systems under classification noise. We begin by establishing the sufficient sample size for a general scheme of multiclass learning under noise using any *element-wise* ERM (defined later). We then propose an **efficient** element-wise ERM

algorithm V-ERM for learning threshold discrete dynamical systems that use $O(1/\epsilon^2 \cdot n^2 \cdot \log{(n/\delta)})$ training samples. Due to space limits, **full proofs appear in the Appendix** A.3.

**A general learning model.** We present a general learning model. Following this, learning discrete dynamical systems becomes a *special case* of this general scheme. This new model follows our learning setting in Section 2 for a *finite* **hypothesis class** $\mathcal{H}'$, with the following generalizations: $(i)$ the training set $\mathcal{T}$ consists of $q$ pairs of $n$-dimensional vectors, denoted by $(\mathcal{W}_j, \hat{\mathcal{W}}_j)$, $j = 1, ..., q$; $(ii)$ the input vector $\mathcal{W}_j \in \{0, 1\}^n$ is drawn from an *unknown* distribution $\mathcal{D}$, where each entry of $\mathcal{W}_j$ is the feature value associated with **an entity**; $(iii)$ the observed label vector $\hat{\mathcal{W}}_j$ in $(\mathcal{W}_j, \hat{\mathcal{W}}_j)$ is an erroneous version of the true label vector $\mathcal{W}'_j$; $(iv)$ the number of labels for each entity is $k \geq 2$. Also see Appendix A.3 for a detailed definition of this general learning model.

### 3.1 ANALYSIS OF ELEMENT-WISE ERM FOR THE GENERAL MODEL

We analyze the number of training samples required by any element-wise ERM for the general multi-class learning problem defined in the previous section; the result extends the sample complexity proof in (Angluin & Laird, 1988) for binary classification.

**Element-wise ERM.** Given a training set $\mathcal{T} = \{(\mathcal{W}_j, \hat{\mathcal{W}}_j)\}_{j=1}^q$, for any hypothesis $h \in \mathcal{H}'$ and entity $i$, $i = 1, ..., n$, we refer to the number of empirical disagreements $\sum_{j=1}^q \mathbb{1}\left(h(\mathcal{W}_j)[i] \neq \hat{\mathcal{W}}_j[i]\right)$ as the **empirical loss** (over $\mathcal{T}$) of $h$ w.r.t entity $i$. Let $\mathcal{A}$ be an *element-wise* ERM algorithm. That is, for any training set $\mathcal{T}$, algorithm $\mathcal{A}$ outputs a hypothesis $h$ from the space $\mathcal{H}'$, such that for every entity $i = 1, ..., n$, the empirical loss of $h$ w.r.t $i$ is **minimized**. We note that for some problems, such an algorithm $\mathcal{A}$ might **not** exist. Therefore, the results in this section are for problems that admit at least one such element-wise ERM.

**Canonical partition of the hypothesis class.** We now to define a partition of the hypothesis class $\mathcal{H}'$. For each $i$th entity, $i = 1, ..., n$, let $\mathcal{P}_i$ be a partition of the hypothesis class $\mathcal{H}'$ into $t_i$ subsets, denoted by $\mathcal{H}'_1, ..., \mathcal{H}'_{t_i}$, such that the following condition holds for each $\mathcal{H}'_\ell$, $\ell = 1, ..., t_i$: *for any training set $\mathcal{T}$, the empirical loss w.r.t. entity $i$ for all hypotheses in $\mathcal{H}'_\ell$ are the same.*

Note that the construction of such a partition is problem-dependent. Clearly, one trivial partition $\mathcal{P}_i$ is of size $|\mathcal{H}'|$, where each subset in the partition consists of just one hypothesis. Given a collection of partitions $\mathcal{P} = \{\{\mathcal{P}_1, ..., \mathcal{P}_n\}\}$ defined above, let $t_{max}(\mathcal{P})$ be the largest $t_i$ over $i = 1, ..., n$.

**Sample complexity.** We now establish the sample complexity of learning under the general framework in Lemma 3.1. We note that the main purpose of Lemma 3.1 is to later derive a sample complexity bound for learning discrete dynamical systems. Nevertheless, this general result may also be of independent interest.

**Lemma 3.1.** *Let $\mathcal{P} = \{\{\mathcal{P}_1, ..., \mathcal{P}_n\}\}$ be a collection of canonical partitions of the hypothesis class $\mathcal{H}'$; $t_{max}(\mathcal{P})$ is the size of the largest partition in $\mathcal{P}$. For any $\epsilon, \delta \in (0, 1)$, and any $\eta_i \leq \bar{\eta} < 1/2$, $i = 1, ..., n$, with a training set of size $q = O\left(\frac{1}{(1-2\bar{\eta})^2} \cdot \frac{1}{\epsilon^2} \cdot n^2 \log(\frac{t_{max}(\mathcal{P}) \cdot n}{\delta})\right)$, any element-wise ERM $\mathcal{A}$ learns a $h \in \mathcal{H}'$ such that with probability at least $1 - \delta$ (over $\mathcal{T} \sim \mathcal{O}^q$), $err_{\mathcal{D}}(h) < \epsilon$.*

**Remark 4.** We note the following: *an element-wise ERM $\mathcal{A}$ in general does **not** minimize the empirical loss over the training set $\mathcal{T}$.* In fact, due to the high probability of receiving a wrong prediction on each sampled data, any true ERM is unlikely to perform well in this case. The second remark is that knowing the exact value of $\bar{\eta}$ is **not** needed in our setting. In particular, one can easily extend the binary search technique introduced in (Angluin & Laird, 1988) (Theorem 3 in (Angluin & Laird, 1988)) to estimate the value of $\bar{\eta}$, such that w.p. at least $1 - \delta$, the estimated $\bar{\eta}$ satisfies $\eta_i \leq \bar{\eta} < 1/2$ for all $i = 1, ..., n$.

### 3.2 AN EFFICIENT VERTEX-WISE ERM FOR LEARNING DISCRETE DYNAMICAL SYSTEMS

Given Lemma 3.1, what remains is to $(i)$ find an actual (efficient) algorithm that is an element-wise ERM and $(ii)$ determine an appropriate partition of the hypothesis class $\mathcal{H}$ for discrete dynamical systems. In this section, we answer these two questions for learning discrete dynamical systems. In particular, we present a simple and efficient algorithm that learns a hypothesis (system) that minimizes the empirical risk w.r.t *each vertex* over the training set $\mathcal{T}$. Further, we show the existence of a canonical partition of $\mathcal{H}$ w.r.t each vertex $v \in \mathcal{V}$, such that the size of each $\mathcal{P}_v$ is at most $\Delta + 2$,

where $\Delta$ is the maximum degree of the graph. Subsequently, we prove that the number of samples needed by the algorithm is $O(1/\epsilon^2 \cdot n^2 \cdot \log(n/\delta))$.

**The algorithm V-ERM.** A system $h$ is inferred as follows: for each vertex $v \in \mathcal{V}(\mathcal{G})$, we learn $\tau_v$ to be the value such that the number of disagreements $\sum_{j=1}^{q} \mathbb{1}\left(h(\mathcal{C}_j)[v] \neq \hat{\mathcal{C}}_j[v]\right)$ over the training set $\mathcal{T}$ is **minimized**. Note that such a $\tau_v$ can be found in polynomial time by iterating over each integer in $[0, \deg_v + 1]$. The pseudocode appears as Algorithm 1 in Appendix A.4.

**The canonical partition.** For each vertex $v$, a canonical partition $\mathcal{P}_v = \{\mathcal{H}_0, ..., \mathcal{H}_{\deg_v+1}\}$ of $\mathcal{H}$ is constructed such that each hypothesis in $\mathcal{H}_\ell$, $\ell = 0, ..., \deg_v + 1$, has the following property: *the threshold of $v$ is $\ell$.* One can verify that, for any training set, the empirical loss w.r.t. vertex $v$ for all hypotheses in $\mathcal{H}_\ell$ are the same. Further, the size of the partition is $\deg_v + 2$. Consequently, $t_{\max}(\mathcal{P}) = \Delta + 2 \leq n + 1$, where $\mathcal{P} = \{\{\mathcal{P}_1, ..., \mathcal{P}_n\}\}$.

Lastly, for our problem of learning threshold discrete dynamical systems, the label for each vertex is either 0 or 1; that is, $k = 2$. By Lemma 3.1, the Theorem follows.

**Theorem 3.2.** *For any $\epsilon, \delta \in (0, 1)$, and any $\eta_v \leq \bar{\eta} < 1/2, v \in \mathcal{V}$, with a training set $\mathcal{T}$ of size $q = O\left(\frac{1}{(1-2\bar{\eta})^2} \cdot \frac{1}{\epsilon^2} \cdot n^2 \cdot \log(\frac{n}{\delta})\right)$, Algorithm 1 (i.e., V-ERM) learns a hypothesis $h \in \mathcal{H}$ such that with probability at least $1 - \delta$ (over $\mathcal{T} \sim \mathcal{O}^q$), $err_{\mathcal{D}}(h) < \epsilon$.*

**Remark 5.** Theorem 3.2 establishes the efficient learnability of threshold discrete dynamical systems under classification noise. Note that the sample complexity bound derived from V-ERM is not optimal, as it follows from the more general result, namely Lemma 3.1. In the next section, we present a more sophisticated algorithm with much lower sample complexity. Nevertheless, it is noteworthy that such a simple algorithm (i.e., V-ERM) can already achieve PAC performance guarantees using only $O(n^2 \log(n))$ noisy samples. The simplicity of this algorithm makes it well-suited for use in practice. In particular, one key property of V-ERM discovered through empirical analysis is that, *the learning accuracy consistently increases as more training samples are provided.* Such a property is *not* observed for the more complex algorithms presented in the next section.

## 4 LEARNING BASED ON VISITING TIMES

One immediate question is whether the sample complexity established in the previous section can be improved. We now answer the question with a problem-dependent analysis and improve the bound to **matches the general upper bound in noise-free scenarios**. Toward this end, we present the algorithm VisRange that uses $O(1/\epsilon \cdot n \log(n/\delta))$ training samples under our high-noise setting, which is only a constant factor larger than the general upper bound for learning under the noise-free scenario (Haussler, 1988). That is, despite the presence of noise, the algorithm achieves efficient learnability without needing many more samples compared to the noise-free scenario. Further, *our established upper bound is only a factor $O(\log n)$ larger than the best-known lower bound* (Aslam & Decatur, 1996). Due to space limit, **all proofs appear in Appendix A.4**.

We first discuss a simplified version of the algorithm for ease of understanding. This simpler algorithm uses $O(1/\epsilon \cdot \Delta n \log(n/\delta))$ training samples, where $\Delta$ is the maximum degree of the graph.

**Visiting a score.** We define the notion of *visiting a score*, which plays a critical role in the algorithm. Let $v$ be a vertex in the graph. Given a configuration $\mathcal{C}$, the **score** of $v$ under $\mathcal{C}$, denoted by $\text{score}(\mathcal{C}, v)$, is the number of state-1 neighbors of $v$ in $\mathcal{C}$. Note that the score of $v$ is in the range $[0, \deg_v + 1]$. For a $\mathcal{C} \sim \mathcal{D}$, we say that a score $s \in [0, \deg_v + 1]$ is **visited** by $\mathcal{C}$ for $v$ if $\text{score}(\mathcal{C}, v) = s$. Subsequently, the **visiting probability** of a score $s$ w.r.t. $v$ is the probability of visiting $s$ over $\mathcal{C} \sim \mathcal{D}$. Given a training set $\mathcal{T} \sim \mathcal{O}^q$, the **visiting time** of a score $s$ is a random variable recording the number of times $s$ got visited by $\mathcal{C}$, summing over all pairs $(\mathcal{C}, \hat{\mathcal{C}}) \in \mathcal{T}$.

**The simplified algorithm VisScore.** Fix a vertex $v \in \mathcal{V}$. At a high level, when the size $q$ of the training set $\mathcal{T}$ is sufficiently large, with probability at least $1 - \delta$, each score with a high visiting probability will be visited a sufficiently large number of times in the training set. We then learn from the majority vote over the output states for each such input score in $\mathcal{T}$.

Formally, let $S$ be the set of scores that are visited at least $q \cdot \epsilon/(2\Delta n)$ times in $\mathcal{T}$. For each score $s \in S$, the algorithm computes the *majority* output state (break ties randomly) of $v$ over the

erroneous successors under input score $s$; let $\ell'_s$ be the majority output state for $s \in S$ over $\mathcal{T}$. Lastly, the threshold of $v$ is learned as $\tau_v = 1 + \max\{s \in S : \ell'_s = 0\}$; $\tau_v = 0$ if $\ell'_s = 1$ for all $s \in S$. This procedure is then performed for each vertex $v \in \mathcal{V}$, which produces a system with all thresholds being inferred. The pseudocode for the algorithm is shown in Algorithm 2 in Appendix A.4.

In Theorem 4.1, we show that Algorithm 2 (i.e., VisScore) uses at most $O(\Delta n \log(n))$ samples to infer all the thresholds. For graphs where the maximum degree is $o(n)$, this bound is already better than the $O(n^2 \log(n))$ bound for the element-wise ERM Algorithm 1 (i.e., V-ERM) in Section 3.

**Theorem 4.1.** *For any $\epsilon, \delta \in (0, 1)$, and any $\eta_v \leq \bar{\eta} < 1/2, v \in \mathcal{V}$, with a training set of size $q = O\left(\frac{1-\bar{\eta}}{(1/2-\bar{\eta})^2} \cdot \frac{1}{\epsilon} \cdot \Delta n \cdot \log(\frac{n}{\delta})\right)$, Algorithm 2 (i.e., VisScore) learns a hypothesis $h \in \mathcal{H}$ such that with probability at least $1 - \delta$ (over $\mathcal{T} \sim \mathcal{O}^q$), we have $err_{\mathcal{D}}(h) < \epsilon$.*

### 4.1 THE ALGORITHM BASED ON VISITING TIMES OF RANGES

In this section, we extend the simplified algorithm (i.e., VisScore ) to our final algorithm VisRange which uses $O(1/\epsilon \cdot n \log(n/\delta))$ training samples. We then show that this bound is tight by comparing it with the general upper and lower bounds from the literature.

**Visiting a range.** We extend the notion of visiting a score (used in VisScore) to *visiting a range of scores*. Let $v$ be a vertex in the graph. Let $R_{s_1,s_2} = [s_1, s_2]$ be a range of scores for $v$, $s_1, s_2 \in \{0, ..., \deg_v + 1\}$, $s_1 \leq s_2$. Note that there are $O(\Delta^2)$ such ranges for each $v$. For a configuration $\mathcal{C} \sim \mathcal{D}$, we say that a range $R_{s_1,s_2}$ is **visited** by $\mathcal{C}$ for $v$ if $\mathrm{score}(\mathcal{C}, v) \in R_{s_1,s_2}$. Similarly, the **visiting probability** of a range $R_{s_1,s_2}$ w.r.t. $v$ is the probability of visiting $R_{s_1,s_2}$ over $\mathcal{C} \sim \mathcal{D}$. Lastly, the **visiting time** of a range $R_{s_1,s_2}$ is a random variable representing the number of times $R_{s_1,s_2}$ got visited by $\mathcal{C}$, summing over all pairs $(\mathcal{C}, \hat{\mathcal{C}}) \in \mathcal{T}$.

**The algorithm VisRange.** Let $S$ be the set of ranges that are visited at least $\epsilon/(2n) \cdot q$ times in $\mathcal{T}$. For each range $(s_1, s_2) \in S$, the algorithm counts the total number of output state-0 and output state-1 over all the erroneous successors with input scores in $[s_1, s_2]$; let $\ell'_{s_1,s_2}$ be the corresponding majority output state for the range $[s_1, s_2]$ over $\mathcal{T}$. Lastly, the threshold of $v$ is learned as $\tau_v = 1 + \max\{s_1 : (s_1, s_2) \in S \text{ and } \ell'_{s_1,s_2} = 0\}$. If $\ell'_{s_1,s_2} = 1$ for all $(s_1, s_2) \in S$, then $\tau_v = 0$. The pseudocode of the algorithm is shown in Algorithm 3 in Appendix A.4. It is clear that the overall algorithm runs in polynomial time. In Theorem 4.2, we show that $O(n \log(n))$ samples are sufficient to learn the system.

**Intuition of the proof.** The algorithm is an extended version of VisScore, where we now care about ranges of scores being visited. Fix a vertex $v \in \mathcal{V}$. Intuitively, when the size $q$ of the training set $\mathcal{T}$ is sufficiently large, with probability at least $1 - \delta$, each range of scores (for $v$) with a high visiting probability would be visited a large number of times. We then learn from the majority vote of the output states of $v$ over all the input scores in each such range with high visiting probabilities.

**Theorem 4.2.** *For any $\epsilon, \delta \in (0, 1)$, and any $\eta_v \leq \bar{\eta} < 1/2, v \in \mathcal{V}$, with a training set of size $q = O\left(\frac{1-\bar{\eta}}{(1/2-\bar{\eta})^2} \cdot \frac{1}{\epsilon} \cdot n \cdot \log(\frac{n}{\delta})\right)$, Algorithm 3 (i.e., VisRange) learns a hypothesis $h \in \mathcal{H}$ such that with probability at least $1 - \delta$ (over $\mathcal{T} \sim \mathcal{O}^q$), we have $err_{\mathcal{D}}(h) < \epsilon$.*

**Remark 6.** The increase in the number of samples required for our VisRange Algorithm is only a factor of $O(1/(1-\eta)^2)$ *compared to the noise-free setting*. We remark that this is a very tight result that one can expect in this domain. Specifically, $(i)$ the presence of $\eta$ is *inevitable*, as it reflects the expected increase in sample complexity when noise level increases; $(ii)$ more importantly, we do **not** incur larger complexity w.r.t the dominant term $n$. This ensures that, in practice, as noise level increases, the increase in the number of samples remains proportional to the effect of noise and does not grow disproportionately. Lastly, we note that the expression $O(1/(1-\eta)^2)$ is common in the sample complexity bound for learning under noise (e.g., (Angluin & Laird, 1988; Laird, 2012).

### 4.2 ALGORITHM SCALABILITY

We remark on the **scalability** of our algorithms to large networks. First, the running time of all the proposed algorithms is $O(n\Delta^2 q)$, where $n$ is the number of vertices, $\Delta$ is the maximum degree, and $q$ is the training set size. More importantly, all algorithms are inherently **parallelizable at**

**vertex level**. This significantly improve the scalability of the model, effectively reducing the time complexity by a factor of $n$, giving a runtime of $O(\Delta^2 q)$.

Further, we have proved that, when the noise level increases, the sufficient number of samples for the algorithm does not grow w.r.t $n$, the dominant term. Consequently, in noisy real-world systems, only a *minimal additional training set* is required to handle increased noise. These features collectively make our algorithms scalable to larger and more complex systems.

**Dependency on $n$.** We now discuss how the dependency on $n$ in the sample complexity (even before parallelization) can be further relaxed. In particular, consider a scenario where only a $\sigma$ number of vertices has unknown interaction functions, and we want to learn these vertices. Then, by the mechanism of the proposed algorithms, our techniques can be naturally extended where the factor $n$ in our sample complexity analysis can be all replaced by $\sigma$. Consequently, when $\sigma$ is low (e.g., only a small fraction of vertices, say $\log(n)$, are to be learned), the corresponding number of samples is significantly reduced, and as $\sigma$ approaches $n$, the bounds approach our bounds.

**Extension to other loss**. We discuss an extension of our algorithm to a natural loss function based on Hamming weights, such as the one given in the the PMAC model (Balcan & Harvey, 2011). In this model, instead of trying to predict the states of all vertices correctly, the notion of a successful prediction is relaxed: *we allow at most $\beta$ fraction of the states of the $n$ vertices to be wrong*. This new setting implies that the Hamming distance between the predicted configuration and the true configuration can be at most $\beta n$. Importantly, our algorithms can be extended to this context without any modification. Further, the resulting sample complexity bound is only increased by an extra multiplicative factor $1/\beta$ and $1/\beta^2$ for Algorithm 1 and Algorithm 2,3, respectively.

### 4.3 Tightness of the sample complexity bound

- **Upper bound.** The work by Haussler (1988) shows that the sample complexity of general PAC learning in the **noise-free** setting is $O\left(\frac{1}{\epsilon} \cdot \log\left(\frac{|\mathcal{H}|}{\delta}\right)\right)$, where $|\mathcal{H}|$ is the size of the hypothesis class. For learning threshold dynamical systems, since $|\mathcal{H}| = O(n^n)$, the corresponding **noise-free** upper bound becomes $O\left(\frac{1}{\epsilon} \cdot n \log\left(\frac{n}{\delta}\right)\right)$.

  In the presence of noise, one would expect the above bound to become higher. Nevertheless, our derived bound under classification noise, $O\left(\frac{1-\bar{\eta}}{(1/2-\bar{\eta})^2} \cdot \frac{1}{\epsilon} \cdot n \cdot \log\left(\frac{n}{\delta}\right)\right)$, is **only a factor** $O(1/(1-\bar{\eta})^2)$ larger than the noise-free upper bound, where the expressions involving the dominant term $n$ remain the same. Our result also matches a general upper bound of $O\left(\frac{1}{(1-2\eta)^2} \cdot \frac{1}{\epsilon} \cdot n \log\left(\frac{n}{\delta}\right)\right)$ by Laird (2012) for PAC learning under classification noise.

  More importantly, **one cannot generally expect efficient PAC learning algorithms to achieve the upper bounds** from Haussler (1988) or Laird (2012) since there exist problems that are not efficiently PAC learnable unless **NP = RP** (Kearns & Vazirani, 1994). Our result (i.e., Theorem 4.2), using VisRange reveals that such a bound indeed holds for efficiently learning threshold dynamical systems.

- **Lower bound.** By a result of Aslam and Decatur (1996), a *general* lower bound on sample complexity for PAC learning under classification noise is $\Omega\left(\frac{1}{(1-2\eta)^2} \cdot \frac{1}{\epsilon} \cdot (\text{Ndim}(\mathcal{H}) + \log\frac{1}{\delta})\right)$, where $\text{Ndim}(\mathcal{H})$ is the Natarajan dimension (Natarajan, 1989) of the class $\mathcal{H}$. For learning threshold systems, it is known that $\text{Ndim}(\mathcal{H}) = n$ (Qiu et al., 2024b). Hence, the lower bound becomes $\Omega\left(\frac{1}{(1-2\eta)^2} \cdot \frac{1}{\epsilon} \cdot (n + \log\frac{1}{\delta})\right)$. Notably, our upper bound in Theorem 4.2 is **only a factor** $O(\log n)$ higher than the general lower bound.

**Remark 7.** With everything in place, we now discuss the advantages of V-ERM (i.e., Algorithm 1) and VisRange (i.e., Algorithm 3) from both theory and practice perspectives:

- **Theory.** With more problem-dependent mechanism and analysis as given for VisRange, one can significantly improve the upper bound (by a multiplicative factor of $n$) of the sufficient training size from the bound derived by V-ERM. Specifically, there exists a problem instance (e.g., a graph $\mathcal{G}$ and a distribution $\mathcal{D}$) such that VisRange requires multiplicative $\Omega(n)$ *fewer* samples than V-ERM to achieve the same error rate. Further, VisRange provides an important (and somewhat surprising) theory insight: despite the presence of classification noise, one can still *efficiently* learn threshold dynamical systems using the *same* number of training samples (to

within only a constant factor) as the noise-free case. Further, the number of required samples is only a factor $O(\log n)$ higher than the lower bound.

- **Practice.** A distinct nature of VisRange is that, one expects to see a drop in the error rate only when the size of the training set $q$ reaches a critical threshold (i.e., a phase transition). Before $q$ reaches that threshold, however, the behavior of the algorithm can be unpredictable as the high-probability guarantee is not yet satisfied. On the other hand, such a critical threshold on the training set size is **not** inherent to V-ERM. In particular, as more training samples are given, the corresponding loss for V-ERM always decreases correspondingly. Thus, V-ERM is the preferred method in practice due to both its simplicity and smoother performance. As a result, we focus on V-ERM for our empirical evaluations, as shown in the next section.

## 5    EXPERIMENTAL EVALUATION

In this section, we present an *exploratory* study on the empirical feasibility of the proposed algorithms, V-ERM and VisRange, on both synthetic and real-world networks (Erdös & Renyi, 1959; Leskovec et al., 2007; Kunegis, 2013). Our goal is to complement our theoretical results by analyzing the empirical behaviors of our algorithms across different model parameters.

**Experimental setup.** The details of the synthetic networks (`Gnp`) and real-world networks are given in Appendix A.5. For each network, we create a target ground-truth system $h^*$ where the thresholds are unknown to the learning algorithm. Under each $h^*$, a training set $\mathcal{T} = \{(\mathcal{C}_j, \hat{\mathcal{C}}_j)\}_{j=1}^q$ is generated such that $(i)$ in each $\mathcal{C}_j$, the state of a vertex is 0 or 1 with the same probability; $(ii)$ $\hat{\mathcal{C}}_j$ is a noisy version of the true successor $\mathcal{C}_j'$ where the noise rate $\eta > 0$ is the same for all vertices. Here, we consider different $\eta$ values ranging from 0.05 to 0.4. Given a training set $\mathcal{T}$, our algorithm then learns a system $h$ by inferring all the thresholds. Lastly, to quantify the solution quality, we sample $1,000$ new configurations and compute the **empirical loss** $\ell$, which is the proportion of the sampled configurations that the learned hypothesis $h$ makes incorrect predictions. The parameter settings are given in Table 1 in the Appendix.

### 5.1    EXPERIMENTAL RESULTS

**Distinct behaviors of the algorithms.** We first examine the behaviors of the algorithms in terms of the loss $\ell$ when more training samples are given. Figure 1 shows the change of loss for V-ERM (i.e., Algorithm 1) and VisRange (i.e., Algorithm 3) for synthetic networks of different sizes $n$. We observe that for larger $n$, VisRange exhibits a phase transition w.r.t. loss when the number of training samples $q$ reaches a critical threshold. Algorithm VisScore (i.e., Algorithm 2) also exhibits a similar phase transition behavior (see Figure 6 in the Appendix). Such phenomena are expected from our theoretical analysis (see Remark 7). In contrast, V-ERM steadily gains in performance when it is given more training samples.

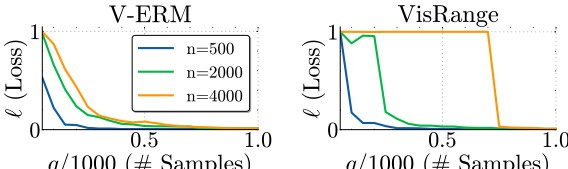

Figure 1: Comparison of V-ERM and VisRange across various network sizes. The average loss is shown after every 50 training samples. Across all the experiments, noise $\eta$ is set to 0.05 and average density $d_{avg}$ is set to 5.

In Figure 1, we further observe that VisRange achieves a loss lower than V-ERM with fewer training samples in the long run for network sizes of 2000 or 4000. Nonetheless, as the network size grows, the required number of training samples for VisRange to reach an acceptable empirical loss also grows. Overall, V-ERM is more suitable for practical purposes, where there is often a trade-off between performance and the number of samples. Since the learning performance of V-ERM steadily improves with increase in the number of samples, such a trade-off can be obtained with fewer training samples with V-ERM compared to VisRange. Overall, although VisRange can promise an improved performance in the long run, V-ERM provides reasonable performance in most practical scenarios. Therefore, the remaining experiments focus on V-ERM.

**Further evaluation of V-ERM.** We perform additional studies on the more applicable algorithm V-ERM. The first evaluation highlights its learning behavior across different noise settings. The results are shown in Figure 2 in two parts, for synthetic networks and real-world networks, respectively.

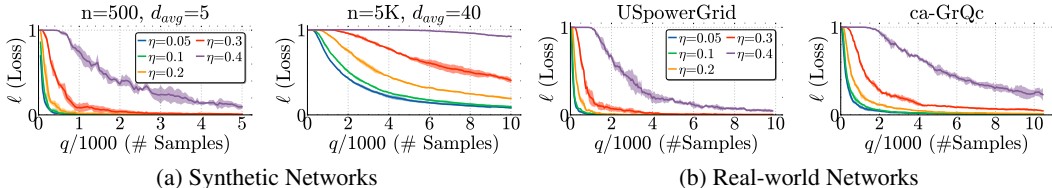

(a) Synthetic Networks        (b) Real-world Networks

Figure 2: Learning curve of V-ERM (Algorithm 1) for various graphs under different noise settings. The shaded region accounts for one standard deviation error.

Figure 2a shows the loss of V-ERM for two synthetic networks under different noise settings. Overall, when the noise rate $\eta$ increases, we see an increase in the number of training samples needed by V-ERM to achieve the same loss. This behavior is expected. The first plot in Figure 2a is for a sparse network with 500 vertices. We find that even under the high noise setting ($\eta = 0.4$), the algorithm performs well. For instance, when $\eta \leq 0.3$, the loss is close to 0 after observing fewer than $3,000$ training samples. The second plot in Figure 2a is for a network with $5,000$ vertices and an average degree of $40$, a larger and much denser network. Even in such case, V-ERM achieves reasonable accuracy for moderate noise levels ($\eta \leq 0.2$). Experimental results for a more comprehensive set of synthetic networks are presented in Figure 8 in Appendix A.5. Similar results for the two real-world networks are shown in Figure 2b. Overall, we observe that V-ERM achieves a high accuracy upon observing samples that are much less than its worst-case theoretical bound (shown in Theorem 3.2).

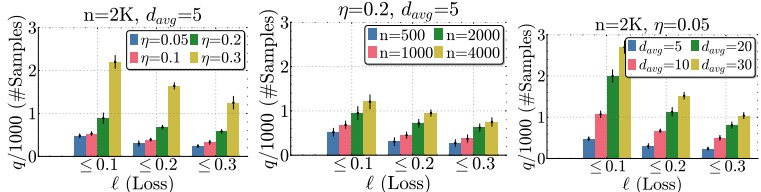

(a) With varying noise    (b) With varying graph size    (c) With varying density

Figure 3: Number of training samples needed for V-ERM for achieving a specified loss for three different variates. Error bars account for two standard deviation errors.

From the observed decreasing pattern of the loss curve, one can conclude that V-ERM will converge towards a zero-loss as it is provided with more training samples. We illustrate this fact using an alternate perspective in Figure 3, where we show the number of training samples required to achieve loss below a certain threshold under three settings: (i) various noise rates (Figure 3a), (ii) various network sizes (Figure 3b), and (iii) various graph densities (Figure 3c). We observe that as the threshold of acceptable loss decreases, the number of required training samples increases. Interestingly, this phenomenon is most sensitive to density, followed by noise and then the network size. Results from a similar experiment on real-world networks are presented in Figure 7 in Appendix A.5.

## 6   Future Work

In this work, we presented efficient algorithms for learning threshold dynamical systems under classification noise. Much work remains to be done for learning discrete dynamical systems under different models of noise. First, it is of interest to investigate whether our algorithms can be extended to other classes of interaction functions such as weighted threshold functions and symmetric functions (Crama & Hammer, 2011). A second direction is to investigate whether the sample complexity bounds can be improved when additional information about a dynamical system (e.g., correct threshold values of some vertices) is available. Finally, it is also of interest to study whether improved learning algorithms can be developed for restricted graph topologies such as planar graphs or intersection graphs.

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

# A APPENDIX

In the Appendix, we present details of the technical results established in the main paper. These include $(i)$ proofs for Lemma 3.1, Theorem 3.2, Theorem 4.1, Theorem 4.2, and $(ii)$ pseudocodes for Algorithm 1 (i.e., V-ERM) and Algorithm 3 (i.e., VisRange). We also include additional experimental results for the algorithms.

## A.1 ADDITIONAL REMARKS

**Remarks on sample complexity analysis.** It is expected that the sample complexity for learning under noise would be higher than the noise-free case, since noise introduces higher variances in the data and higher uncertainty in the hypothesis space. Further, noise leads to misleading patterns in the training process. For a simple example, in the training set, we often observe that under the same input to a vertex v's (deterministic) interaction function, the outputs are sometimes 0 and sometimes 1 (due to noise). Such uncertainties require more training samples for an algorithm to unpack useful information about the true hypothesis.

**Impact of the theoretical findings.** Below we address additional impact of the findings from both theoretical and practical perspectives.

1. **Theory.** Existing work on learning threshold discrete dynamical systems focuses on the noise-free setting; the problem of efficient learning under noise has remained *open*. We fill this gap and establish that one can efficiently learn threshold discrete dynamical systems under high classification noise. Further, our proposed algorithms are strongly noise-tolerant, with only a marginal increase in the number of samples used in comparison with the noise-free case. Overall, we establish a theoretical foundation for learning threshold discrete dynamical systems in the presence of noise.

2. **Practice.** Our results provides practical methodologies on learning real-world large-scale complex systems (e.g., social, multi-agent, infrastructure systems).

   **Practical applications**. As discussed in the paper, discrete dynamical systems are widely used to model dynamic processes in various fields such as biology, social sciences, and network analysis. Our results provide efficient techniques for robust modeling and inference in these domains, under the realistic setting where training data is *noisy* (which is often the case in fields like social science). This robustness w.r.t noise is crucial for practical applications when $(i)$ perfect data are rare, and still, $(ii)$ reliable decisions are needed.

   **Training data.** Despite the presence of noise, the increase in the required number of samples for our proposed algorithm is marginal. As a result, even when the noise level rises in real-world applications, practitioners do not need to excessively increase data collection efforts while maintaining model accuracy. This efficiency is particularly beneficial in fields where data collection is expensive, such as social science which often involves extensive surveying and field research.

   **Model simplicity.** The algorithms are in the classic ML domain, and they are inherently simpler and interpretable compared to deep learning methods. This simplicity translates into more straightforward implementation and less intricate engineering effort. As a result, practitioners can quickly deploy these algorithms in real-world applications. Further, because the learning process employs *gradient-free* optimization, the extensive computation associated with back-propagation is not a bottleneck for the algorithms, making them accessible to a broader range of users with limited computational resources. These are also reflected in our code and experimental evaluation which use only standard (single thread) CPUs. Consequently, the transparent training process can easily be interpreted since it involves only a small number of parameters.

   **Scalability.** As discussed in the main paper, our proposed algorithms can learn the interaction function for each vertex independently. Therefore, the algorithms can be easily scaled across batches of vertices given the necessary computational resources, making the algorithms suitable for deployment in settings where analysis over large-scale networks is required.

## A.2 ADDITIONAL INFORMATION FOR SECTION 2

**The settings of existing studies on learning dynamical systems**

Our setting for learning discrete dynamical systems (see Section 2 in the main paper) aligns with the line of existing research. Here, we present the detailed setting of a few illustrative papers on learning discrete dynamical systems. These works are also cited in our main paper.

All the existing work presented below considers the following setting: 1. The vertex state is **binary**; 2. The update scheme is **synchronous**; 3. The time-scale is **discrete**; 4. The interaction functions are either threshold functions, susceptible-infected, or independent cascade.

`ICML-2024` (Qiu et al., 2024b); `AAAI-2022` (Conitzer et al., 2022); `ICML-2022` (Rosenkrantz et al., 2022); `ICML-2021` (Chen et al., 2021); `ICML-2021` (Dawkins et al., 2021); `ICML-2021` (Wilinski & Lokhov, 2021); `NeurIPS-2020` (Li et al., 2020); `ICML-2020` (Conitzer et al., 2020); `ICML-2019` (Adiga et al., 2019); `NeurIPS-2016` (He et al., 2016); `NeurIPS-2015` (Narasimhan et al., 2015).

**A pictorial example of a discrete dynamical system**

We present a toy example of the evolution of a threshold discrete dynamical system. The goal of this figure is to assist readers in understanding the formal definitions presented in Section 2 of the main paper. For large-scale realistic discrete dynamical systems used in the real world, we refer readers to the following references: (Battiston et al., 2020; Ji et al., 2017; Lum et al., 2014; Sneddon et al., 2011; Schelling, 2006; Laubenbacher & Stigler, 2004; Kauffman et al., 2003).

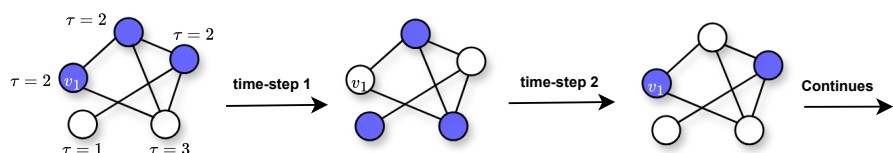

Figure 4: The evolution of an example threshold dynamical system with 5 vertices. The threshold value of each vertex is shown in the figure. Here, we present system updates over two time-steps, where vertices in state-1 are highlighted in blue. For instance, the threshold of vertex $v_1$ is 2. In the first configuration, $v_1$ has only one neighbor in state-1, which is less than its threshold. Therefore, its state gets updated to 0 (shown in the second configuration) after time step 1. In the second configuration, $v_1$ has two neighbors of type-1, which satisfies its threshold value. Thus, its state gets updated to 1 after time-step 2.

## A.3 ADDITIONAL INFORMATION FOR RESULTS IN SECTION 3

**The general learning model**

We now present the details of the general learning model stated in Section 3. Following this, learning discrete dynamical systems becomes a *special case* of this general scheme. The new scheme follows our learning setting in Section 2 for a *finite* hypothesis class $\mathcal{H}'$, with the following generalizations:

- **Training set.** The training set contains *pairs* of $n$-dimensional vectors. For each pair, the input vector represents the features of $n$ entities (e.g., vertices); their corresponding labels are computed from the input features *collectively*.
  Formally, a training set $\mathcal{T}$ of size $q$ consists of $q$ pairs of $n$-dimensional vectors, denoted by $(\mathcal{W}_j, \hat{\mathcal{W}}_j)$, $j = 1, ..., q$. For each $(\mathcal{W}_j, \hat{\mathcal{W}}_j) \in \mathcal{T}$, the input vector $\mathcal{W}_j \in \{0, 1\}^n$ is drawn from an *unknown* distribution $\mathcal{D}$, where each entry of $\mathcal{W}_j$ is the feature value associated with an entity. Let $\mathcal{W}'_j$ denote the *true* label for $\mathcal{W}_j$, computed by some unknown ground truth labeling function (i.e., hypothesis) $h^* \in \mathcal{H}'$. Here, each $\mathcal{W}'_j[i] \in \{0, ..., k - 1\}$ is the label for the $i$th entity, $i = 1, ..., n$, where there are $k \geq 2$ possible labels. We do not restrict the actual form of $h^*$.

- **Noise process**. The observed label vector $\hat{\mathcal{W}}_j$ in $(\mathcal{W}_j, \hat{\mathcal{W}}_j)$ is an erroneous version of $\mathcal{W}'_j$. In particular, for each entity $i = 1, ..., n$ and for some $0 \leq \eta_i < 1/2$, $(i)$ with probability $1 - \eta_i$, the value of $\hat{\mathcal{W}}_j[i] = \mathcal{W}'_j[i]$; $(ii)$ with probability $\eta_i$, $\hat{\mathcal{W}}_j[i]$ is a label in $\{0, ..., k-1\} \setminus \{\mathcal{W}'_j[i]\}$ that is *different* from the true label $\mathcal{W}'_j[i]$, chosen uniformly at random from the $k-1$ labels. Let $\bar{\eta} < 1/2$ be an upper bound on the error terms.

The goal is to search for a hypothesis $h \in \mathcal{H}'$ such that, when given a new feature vector $\mathcal{W} \sim \mathcal{D}$, $h$ predicts the resulting true label vector $\mathcal{W}'$ under the PAC guarantee.

**Detailed proof of Lemma** 3.1.

Recall that Lemma 3.1 establishes an upper bound on the number of noisy training samples needed by any element-wise ERM for learning under the general learning model.

**Lemma 3.1** *Let $\mathcal{P} = \{\mathcal{P}_1, ..., \mathcal{P}_n\}$ be a collection of partitions of the hypothesis class $\mathcal{H}'$; $t_{max}(\mathcal{P})$ is the size of the largest partition in $\mathcal{P}$. For any $\epsilon, \delta \in (0, 1)$, and any $\eta_i \leq \bar{\eta} < 1/2$, $i = 1, ..., n$, with a training set of size*

$$q = O\left(\frac{1}{(1 - 2\bar{\eta})^2} \cdot \frac{1}{\epsilon^2} \cdot n^2 \log(\frac{t_{\max}(\mathcal{P}) \cdot n}{\delta})\right)$$

*any element-wise ERM $\mathcal{A}$ learns a hypothesis $h \in \mathcal{H}'$ such that with probability at least $1 - \delta$ (over $\mathcal{T} \sim \mathcal{O}^q$), $err_{\mathcal{D}}(h) < \epsilon$.*

**Proof.** We show that a training set of size

$$q = \left\lceil \frac{8\bar{\eta} \cdot n^2 + \epsilon(1 - (2 - k') \cdot \bar{\eta}) \cdot n}{\epsilon^2 \cdot (1 - (2 - k') \cdot \bar{\eta})^2} \cdot \ln\left(\frac{2t_{\max}(\mathcal{P}) \cdot n}{\delta}\right) \right\rceil \tag{1}$$

is sufficient to establish the $(\epsilon, \delta)$-PAC guarantee, where $k' = (k-2)/(k-1)$.

Let

$$err_{\mathcal{D}}(h, i) = \Pr_{\mathcal{W} \sim \mathcal{D}}[h(\mathcal{W})[i] \neq h^*(\mathcal{W})[i]]$$

be the probability (over $\mathcal{W} \sim \mathcal{D}$) of a hypothesis $h$ makes a wrong prediction on the state of the entity $i = 1, ..., n$. We say that a hypothesis $h \in \mathcal{H}$ is $\gamma$-**bad** w.r.t entity $i$ if $err_{\mathcal{D}}(h, i) > \gamma$.

We show the following result.

**Claim A.1.** The probability (over $\mathcal{T} \sim \mathcal{O}^q$) that the learned hypothesis $h$ is $\epsilon/n$-bad w.r.t. at least one entity $i = 1, .., n$ is at most $\delta$

Note that by the above claim,

$$\Pr_{\mathcal{T} \sim \mathcal{O}^q}[err_{\mathcal{D}}(h) > \epsilon] \leq \delta \tag{2}$$

since if $err_{\mathcal{D}}(h) > \epsilon$ for the learned $h$, it must be the case that $err_{\mathcal{D}}(h, i) > \epsilon/n$ for at least one entity $i = 1, .., n$. We thus have

$$\Pr_{\mathcal{T} \sim \mathcal{O}^q}[err_{\mathcal{D}}(h) > \epsilon] \leq \Pr_{\mathcal{T} \sim \mathcal{O}^q}[err_{\mathcal{D}}(h, i) > \epsilon \text{ for at least one } i = 1, ..., n] \leq \delta$$

and Lemma 3.1 thus follows. The rest of the proof focuses on showing that Claim A.1 holds under the training size $q$ given in Eq 1.

First, **fix an entity** $i \in \{1, ..., n\}$. For a noisy training data point $(\mathcal{W}, \hat{\mathcal{W}}) \sim \mathcal{O}$ provided by the oracle, let $e\hat{r}r_{\mathcal{O}}(h, i)$ be the probability of $h(\mathcal{W})$ disagrees with the state of $i$ in $\hat{\mathcal{W}}$ returned from the erroneous oracle $\mathcal{O}$. Formally,

$$e\hat{r}r_{\mathcal{O}}(h, i) = \Pr_{(\mathcal{W}, \hat{\mathcal{W}}) \sim \mathcal{O}}[h(\mathcal{W})[i] \neq \hat{\mathcal{W}}[i]]$$

Note that for the ground-truth hypothesis $h^*$, it holds that $e\hat{r}r_{\mathcal{O}}(h^*, i) = \eta_i$ since the disagreement only happens when an erroneous state is returned by the oracle $\mathcal{O}$. Similarly, for any hypothesis $h$ that is $\epsilon/n$-bad w.r.t. the entity $i$, one can easily verify that

$$e\hat{r}r_{\mathcal{O}}(h, i) = (1 - err_{\mathcal{D}}(h, i)) \cdot \eta_i + err_{\mathcal{D}}(h, i) \cdot (1 - \eta_i) + err_{\mathcal{D}}(h, i) \cdot \eta_i \cdot k' \tag{3}$$

$$\geq \eta_i + \frac{\epsilon}{n} \cdot (1 - (2 - k')\bar{\eta})$$

where $k' = (k-2)/(k-1)$.

Let

$$z = \frac{\epsilon}{n}(1 - (2 - k')\bar{\eta})$$

be the lower bound on the difference of the error probability (over $(\mathcal{W}, \hat{\mathcal{W}}) \sim \mathcal{O}$) between $h^*$ and any hypothesis $h$ that is $\epsilon/n$-bad w.r.t. entity $i$.

We now **fix a hypothesis** $h'$ that is $\epsilon/n$-bad w.r.t. entity $i$. Note that Claim A.1 trivially holds if no such $h'$ exists. Let $X(h', i, \mathcal{T})$ be the random variable representing the number of disagreements over $\mathcal{T}$ between the states of $i$ predicted by $h'$ and the states of $i$ returned by the erroneous oracle $\mathcal{O}$. That is, $X(h', i, \mathcal{T})$ is number of samples $(\mathcal{W}_j, \hat{\mathcal{W}}_j)$ in $\mathcal{T}$ such that $\hat{\mathcal{W}}_j[i] \neq h(\mathcal{W}_j)[i]$.

Note that if this hypothesis $h'$ is returned by our element-wise ERM $\mathcal{A}$, the empirical loss of $h'$ must be the minimum over all hypotheses in $\mathcal{H}$, including the true hypothesis $h^*$. Thus, at least one of the following two events must occur: $(I)$ $X(h^*, i, \mathcal{T}) > \eta_i q + z/2 \cdot q$; $(II)$ $X(h', i, \mathcal{T}) \leq \eta_i q + z/2 \cdot q$.

Observe that

$$\mathbb{E}[X(h^*, i, \mathcal{T})] = \eta_i \cdot q, \quad \mathbb{E}[X(h', i, \mathcal{T})] \geq (\eta_i + z)q$$

where the expectation is taken over $\mathcal{T} \sim \mathcal{O}^q$. By Chernoff,

$$\Pr_{\mathcal{T} \sim \mathcal{O}^q}[X(h', i, \mathcal{T}) \leq \eta_i q + z/2 \cdot q] = \Pr_{\mathcal{T} \sim \mathcal{O}^q}[X(h', i, \mathcal{T}) \leq (1 - \frac{z}{2(\eta_i + z)}) \cdot (\eta_i + z)q]$$

$$\leq \exp\left(-\frac{1}{8} \cdot q \cdot \frac{z^2}{\eta_i + z}\right) \tag{4}$$

One can then verify that, for the value of $q$ specified in Eq 1, we have

$$\exp\left(-\frac{1}{8} \cdot q \cdot \frac{z^2}{\eta_i + z}\right) < \frac{1}{2} \cdot \frac{\delta}{t_{\max}(\mathcal{P})n}$$

and thus by Ineq 4,

$$\Pr_{\mathcal{T} \sim \mathcal{O}^q}[X(h', i, \mathcal{T}) \leq \eta_i q + z/2 \cdot q] < \frac{1}{2} \cdot \frac{\delta}{t_{\max}(\mathcal{P})n} \tag{5}$$

Similarly, one can easily verify that for the same $q$,

$$\Pr_{\mathcal{T} \sim \mathcal{O}^q}[X(h^*, i, \mathcal{T}) > \eta_i q + z/2 \cdot q] < \frac{1}{2} \cdot \frac{\delta}{n} \tag{6}$$

Recall that the hypothesis class $\mathcal{H}'$ admits a collection of partitions $\mathcal{P} = \{\mathcal{P}_1, ..., \mathcal{P}_n\}$. Under each partition $\mathcal{P}_j \in \mathcal{P}$, the empirical loss (under any training set $\mathcal{T}$) w.r.t the entity $j$ for all hypotheses in each subset of $\mathcal{H}'$ are the same. Suppose the event "$X(h', i, \mathcal{T}) \leq \eta_i q + z/2 \cdot q$" occurs for $h'$, then this event *must* also occurred for all other hypotheses that are in the same subset with $h'$ under $\mathcal{P}_i$.

Since the size of the partition $\mathcal{P}_i$ is $t_i \leq t_{max}$, by Ineq 5, it follows that the probability (over $\mathcal{T} \sim \mathcal{O}^q$) of the event "$X(h', v, \mathcal{T}) \leq \eta_i q + z/2 \cdot q$" occurs for at least one $\epsilon/n$-bad hypothesis $h'$ in the space $\mathcal{H}$ is at most $1/2 \cdot \delta/n$.

Combining Ineq 5 and Ineq 6, it follows that the probability (over $\mathcal{T} \sim \mathcal{O}^q$) of either

**Event I**: "$X(h^*, i, \mathcal{T}) > \eta_i q + z/2 \cdot q$"

or

**Event II**: "$X(h', i, \mathcal{T}) \leq \eta_i q + z/2 \cdot q$ for at least one $\epsilon/n$-bad hypothesis $h'$"

occurs is at most $\delta/n$.

Lastly, let $h$ denote the hypothesis returned by the element-wise ERM $\mathcal{A}$. Note that $h$ is $\epsilon/n$-bad (i.e., $err_{\mathcal{D}}(h, i) > \epsilon/n$) w.r.t $i$ only if either event **I** or event **II** (or both) happened. Thus, we have

$$\Pr_{\mathcal{T} \sim \mathcal{O}^q}[err_{\mathcal{D}}(h, i) > \epsilon/n] \leq \frac{\delta}{n} \tag{7}$$

Subsequently,

$$\Pr_{\mathcal{T} \sim \mathcal{O}^q}[err_{\mathcal{D}}(h, i) > \epsilon/n \text{ for at least one entity } i = 1, ..., n] \leq \delta \tag{8}$$

Claim A.1 follows. This concludes the proof. ∎

**Pseudocode of Algorithm 1** Here, we present the pseudocode of Algorithm 1 (i.e., V-ERM).

---

**ALGORITHM 1:** `Vertex-wise ERM` (V-ERM)

**Input** : A training set $\mathcal{T}$; graph $\mathcal{G}$
**Output:** A system $h$ from $\mathcal{H}$

1 **for** $v \in \mathcal{V}(\mathcal{G})$ **do**
2     **for** $\tau = 0, 1, ..., \deg_v + 1$ **do**
3        $h_\tau \leftarrow$ a system where the threshold of $v$ is $\tau$
4        $s_\tau \leftarrow \sum_{(\mathcal{C}, \hat{\mathcal{C}}) \in \mathcal{T}} \mathbb{1}\left(h_\tau(\mathcal{C})[v] \neq \hat{\mathcal{C}}[v]\right)$
5     **end**
6     In $h$, set $\tau_v \leftarrow \arg\min_\tau\{s_\tau\}$             // The threshold of $v$ in $h$
7 **end**
8 **return** $h$

---

**Detailed proof of Theorem 3.2**

Theorem 3.2 establishes the sufficient number of training samples for Algorithm 1 on learning threshold discrete dynamical systems. This result is an implication of Lemma 3.1. In particular, one can define a partition of the hypothesis class $\mathcal{H}$ as follows. For each vertex $v \in \mathcal{V}$, consider a partition $\mathcal{P}_v = \{\mathcal{H}_0, ..., \mathcal{H}_{\deg_v + 1}\}$ of $\mathcal{H}$ such that each hypothesis in $\mathcal{H}_\ell$, $\ell = 1, ..., \deg_v + 1$, has the property: *the threshold of $v$ is $\ell$*. One can easily verify that, for any training set $\mathcal{T}$, the empirical loss w.r.t. vertex $v$ for all hypotheses in $\mathcal{H}_\ell$ are the same. Further, note that the size of the partition $t_v = \deg_v + 2$. Subsequently, the value $t_{\max}(\mathcal{P}) = \Delta + 2$ where $\mathcal{P} = \{\mathcal{P}_1, ..., \mathcal{P}_n\}$. By Lemma 3.1, it follows that a training set of size

$$q = \left\lceil \frac{8\bar{\eta} \cdot n^2 + \epsilon(1 - 2\bar{\eta}) \cdot n}{\epsilon^2 \cdot (1 - 2\bar{\eta})^2} \cdot \ln\left(\frac{3\Delta \cdot n}{\delta}\right) \right\rceil$$

is sufficient. Since $\Delta < n$, the theorem follows.

**Theorem 3.2** *For any $\epsilon, \delta \in (0, 1)$, and any $\eta_v \leq \bar{\eta} < 1/2, v \in \mathcal{V}$, with a training set of size*

$$q = O\left(\frac{1}{(1 - 2\bar{\eta})^2} \cdot \frac{1}{\epsilon^2} \cdot n^2 \cdot \log(\frac{n}{\delta})\right)$$

*Algorithm 1 (i.e., (i.e., V-ERM)) learns a hypothesis $h \in \mathcal{H}$ such that with probability at least $1 - \delta$ (over $\mathcal{T} \sim \mathcal{O}^q$), $err_{\mathcal{D}}(h) < \epsilon$.*

A.4 ADDITIONAL INFORMATION FOR RESULTS IN SECTION 4

**Pseudocode for Algorithm 2.** We present the pseudocode for Algorithm 2 VisScore on learning based on *visiting times of scores*.

---

**ALGORITHM 2:** `Visiting Scores` (**VisScore**)

**Input** : A training set $\mathcal{T}$; graph $\mathcal{G}$
**Output:** A system $h$

1   $q \leftarrow |\mathcal{T}|$                    `// Size of the training set`
2   **for** $v \in \mathcal{V}(\mathcal{G})$ **do**
3      $\lambda_s \leftarrow 0, s = 0, ..., \deg_v + 1$         `// The hitting time of score`
4      $a_s, b_s \leftarrow 0, s = 0, ..., \deg_v + 1$   `// Count the number of output state-0's`
         `and output state-1's under input score s`
5      **for** $(\mathcal{C}, \hat{\mathcal{C}}) \in \mathcal{T}$ **do**
6          $s \leftarrow \text{score}(\mathcal{C}, v)$
7          $\lambda_s \leftarrow \lambda_s + 1$
8          $a_s \leftarrow a_s + 1$ **if** $\hat{C}[v] == 0$; **else** $b_s \leftarrow b_s + 1$
9      **end**
10      $S \leftarrow \emptyset$
11      **for** $s = 0, ..., \deg_v + 1,$ **do**
12          **if** $\lambda_s \geq \frac{\epsilon}{2\Delta n} \cdot q$ **then**
13              $\ell'_s \leftarrow 0$ **if** $a_s > b_s$; **else** $\ell'_s \leftarrow 1$    `// Majority voting on the correct`
                `output state of v under input score s`
14              $S \leftarrow S \cup \{s\}$
15          **end**
16      **end**
17      **if** $\exists\, s \in S$ *s.t.* $\ell'_s = 0$ **then**
18          In $h$, set $\tau_v \leftarrow 1 + \max\{s : s \in S, \ell'_s = 0\}$ `// The learned threshold of v`
19      **end**
20      **else**
21          In $h$, set $\tau_v \leftarrow 0$
22      **end**
23 **end**
24 **return** $h$

---

**Detailed proof of Theorem 4.1**

In Theorem 4.1, we prove the sufficient number of training samples needed by Algorithm 2.

**Theorem 4.1** *For any $\epsilon, \delta \in (0, 1)$, and any $\eta_v \leq \bar{\eta} < 1/2, v \in \mathcal{V}$, with a training set of size*

$$q = O\left(\frac{1 - \bar{\eta}}{(1/2 - \bar{\eta})^2} \cdot \frac{1}{\epsilon} \cdot \Delta n \cdot \log(\frac{n}{\delta})\right)$$

*Algorithm 2 (i.e., **VisScore**) learns a hypothesis $h \in \mathcal{H}$ such that with probability at least $1 - \delta$ (over $\mathcal{T} \sim \mathcal{O}^q$), we have $err_{\mathcal{D}}(h) < \epsilon$.*

**Proof.** We prove that a training set of size

$$q = 4 \cdot \frac{1 - \bar{\eta}}{(1/2 - \bar{\eta})^2} \cdot \frac{\Delta n}{\epsilon} \cdot \ln\left(\frac{4\Delta n}{\delta}\right) \tag{9}$$

is sufficient for Algorithm 2 to guarantee the $(\epsilon, \delta)$-PAC bound. Recall that $\Delta < n$ is the maximum degree of the underlying graph. From a high level, we show that, when $q$ is sufficiently large, with probability at least $1 - \delta$ over $\mathcal{T} \sim \mathcal{D}^q$, each score with a relatively "high" visiting probability will be visited a sufficiently large number of times. We then take the majority output state of $v$ over the erroneous successors under each such score to be the correct output state and subsequently infer the threshold $v$.

As shown in Algorithm 2, we learn the set of thresholds in a vertex-wise manner. **Fix a vertex** $v \in \mathcal{V}$. For each of $v$'s possible scores, denoted by $s = 0, ..., \deg_v + 1$, we say that a score $s$ is $\epsilon$-**important** w.r.t. $v$ if its visiting probability (over $\mathcal{C} \sim \mathcal{D}$) is at least $\epsilon$. Recall that the visiting probability of a score $s$ in terms of $v$ is the probability of sampling a configuration $\mathcal{C} \sim \mathcal{D}$ such that the score of $v$ under $\mathcal{C}$ is $s$.

Now, we **fix a score** $s$ that is $(\epsilon/(\Delta n))$-important w.r.t $v$. Note that at least one score in $\{0, ..., \deg_v\}$ is $(\epsilon/(\Delta n))$-important w.r.t $v$. Let $X(s, \mathcal{T})$ be the random variable (over $\mathcal{T} \sim \mathcal{O}^q$) representing the *visiting times* of the score $s$ in the training set $\mathcal{T}$. That is, $X(s, \mathcal{T})$ is the number of $(\mathcal{C}, \hat{\mathcal{C}}) \in \mathcal{T}$ such that the score of $v$ under $\mathcal{C}$ is $s$.

Importantly, since $s$ is $(\epsilon/(\Delta n))$-important, the expected value of $X(s, \mathcal{T})$ satisfies

$$\mathbb{E}[X(s, \mathcal{T})] \geq \frac{\epsilon}{\Delta n} \cdot q$$

Recall that in Algorithm 2, we only learn from the scores whose visiting time in $\mathcal{T}$ is at least

$$t = \frac{1}{2} \cdot \frac{\epsilon}{\Delta n} \cdot q$$

Ideally, every $(\epsilon/(\Delta n))$-important score is visited at least $t = 1/2 \cdot \epsilon/(\Delta n) \cdot q$ when $q$ is sufficiently large. Specifically, by tail bound, one can verify that

$$\Pr_{\mathcal{T} \sim \mathcal{O}^q}[X(s, \mathcal{T}) < t] \leq \exp\left(-\frac{1}{8} \cdot \frac{\epsilon}{\Delta n} \cdot q\right) \tag{10}$$

where the event "$X(s, \mathcal{T}) < t$" occurring for the $(\epsilon/(\Delta n))$-important score $s$ is undesirable. It then follows that

$$\Pr_{\mathcal{T} \sim \mathcal{Q}^q}[X(s, \mathcal{T}) < t \text{ for at least one } \epsilon/(\Delta n)\text{-important score } s] \tag{11}$$

$$\leq 2\Delta \cdot \exp\left(-\frac{1}{8} \cdot \frac{\epsilon}{\Delta n} \cdot q\right)$$

there there are at most $\Delta$ scores for vertex $v$. For clarity, we define the above (bad) event:

**Event I:** $X(s, \mathcal{T}) < 1/2 \cdot \frac{\epsilon}{\Delta n} \cdot q$ for at least one $\epsilon/(\Delta n)$-important score $s$

One can then verify that, when the size of the training set $q$ satisfies Eq 9, Event I happens with probability (over $\mathcal{T} \sim \mathcal{O}^q$) at most $\delta/(2n)$.

Another desirable property that Algorithm 2 utilizes is that, for sufficiently large $q$, when a score $s$ is visited enough number of times (i.e., at least $t = 1/2 \cdot \epsilon/(\Delta n) \cdot q$ times) in $\mathcal{T}$, the *majority output state* of $v$ over the erroneous successor under the input score $s$ is the *true state* of $v$ in an error-free successor under the same input score $s$. We now prove this.

We **fix a score** $s$ of $v$ that got visited at least $t = (1/2) \cdot \epsilon/(\Delta n) \cdot q$ times in $\mathcal{T}$. Note that such a score must exist. Let $Q(s, T)$ be the visiting times for $s$ under $\mathcal{T}$; $Q(s, \mathcal{T}) \geq t$. Let $\ell_s$ be the correct output state of $v$ under the input score $s$. That is, when there are no errors, if the input to $v$'s interaction function is $s$, then the state of $v$ returned by the ground-truth system $h^*$ is $\ell_s$.

Let $Y(\ell_s, s, \mathcal{T})$ be the number of times that $\ell_s$ appears as the output state of $v$ over the erroneous successors in the training set $\mathcal{T}$, under the input score $s$. Note that

$$\mathbb{E}[Y(\ell_s, s, \mathcal{T})] \geq (1 - \bar{\eta}) \cdot Q(s, T) \geq (1 - \bar{\eta}) \cdot t$$

where $\bar{\eta} < 1/2$ is the upper bound on the error terms over all vertices. Ideally, we should have $Y(\ell_s, s, \mathcal{T}) > 1/2 \cdot Q(s, T)$, that is, the state $\ell_s$ wins is the majority vote. By Chernoff, one can verify that:

$$\Pr_{\mathcal{T} \sim \mathcal{O}^q}[Y(\ell_s, s, \mathcal{T}) \leq \frac{1}{2} \cdot Q(s, T)] \tag{12}$$

$$= \Pr_{\mathcal{T} \sim \mathcal{O}^q}[Y(\ell_s, s, \mathcal{T}) \leq (1 - \frac{(1 - \bar{\eta}) - 1/2}{1 - \bar{\eta}}) \cdot ((1 - \bar{\eta}) \cdot Q(s, T))]$$

$$\leq \exp\left(-\frac{1}{2} \cdot \left(\frac{(1 - \bar{\eta}) - 1/2}{1 - \bar{\eta}}\right)^2 \cdot (1 - \bar{\eta}) \cdot Q(s, T)\right)$$

$$\leq \exp\left(-\frac{1}{2} \cdot \left(\frac{(1 - \bar{\eta}) - 1/2}{1 - \bar{\eta}}\right)^2 \cdot (1 - \bar{\eta}) \cdot \frac{\epsilon q}{2\Delta n}\right)$$

where the event "$Y(\ell_s, s, \mathcal{T}) \leq \frac{1}{2} \cdot Q(s, T)$" occurring for the label $\ell_s$ is undesirable. It follows that

$$\Pr{}_{\mathcal{T} \sim \mathcal{O}^q}[Y(\ell_s, s, \mathcal{T}) \leq \frac{1}{2} \cdot Q(s, T) \text{ for at least one score } s \text{ with visiting time at least } t] \quad (13)$$

$$\leq 2\Delta \cdot \exp\left(-\frac{1}{2} \cdot \left(\frac{(1 - \bar\eta) - 1/2}{1 - \bar\eta}\right)^2 \cdot (1 - \bar\eta) \cdot \left(\frac{1}{2}\frac{\epsilon}{\Delta n}q\right)\right)$$

Let Event II be the above bad event:

**Event II:** $Y(\ell_s, s, \mathcal{T}) \leq 1/2 \cdot Q(s, T)$ for at least one score $s$ with visiting time at least $t$

One can verify that for a training set of size $q$ shown in Eq 9, we have that Event II happens with probability (over $\mathcal{T} \sim \mathcal{O}^q$) at most $\delta/(2n)$.

Let $h$ be the hypothesis returned by Algorithm 2. We say that hypothesis $h$ is $\epsilon/n$-**good** w.r.t $v$ if $\Pr_{\mathcal{C} \sim \mathcal{D}}[h(\mathcal{C})[v] \neq h^*(\mathcal{C})[v]] < \epsilon/n$. We remark that:

**Claim A.2.** If **both** Event I and Event II do **not** occur under a training set $\mathcal{T}$, then the learned $h$ must be $\epsilon/n$-good w.r.t $v$.

To see this, suppose that both Event I and Event II do **not** occur. Then, the following are true simultaneously:

$(i)$ Every $\epsilon/(\Delta n)$-important score got visited at least $t$ times.

$(ii)$ For any scores that got visited at least $t$ times, the majority voting scheme (over $\mathcal{T}$) gives the correct output state of $v$ under the input score $s$ to $v$'s interaction function.

Let $S$ be the set of scores that got visited at least $t$ times. Note that $S$ includes all the $(\epsilon/(\Delta n))$-important scores, plus possibly some other scores.

Since both Event I and Event II do **not** occur, $h$ learns the correct output state of $v$ under each input score in $S$. Let $S' \subseteq S$ be the subset of scores such that the learned output state of $v$ is 0. If $S'$ is an empty set, we then have $\tau_v = 0$ as stated in Algorithm 2. On the other hand, suppose $S'$ contains at least one score. As shown in Algorithm 2, we then set

$$\tau_v = \max{}_{s \in S'}\{s\} + 1$$

For either case, one can easily verify that $h$ would **not** make a wrong prediction on the output state of $v$ when seeing any scores in $S$ as an input. Consequently,

*For a $\mathcal{C} \sim \mathcal{D}$, the hypothesis $h$ can make a **wrong** prediction on the output state of $v$ only when the input score (under $\mathcal{C}$) is **not** in $S$.*

Note that any score not in $S$ is **not** $(\epsilon/(\Delta n))$-important w.r.t. $v$. That is, the probability of visiting a score $s \notin S$ of $v$ under $\mathcal{C} \sim \mathcal{D}$ is less that $\epsilon/(\Delta n)$. Since there are at most $\Delta$ such "bad" scores, we have $\Pr_{\mathcal{C} \sim \mathcal{D}}[h(\mathcal{C})[v] \neq h^*(\mathcal{C})[v]] < \epsilon/n$. This concludes the claim.

Lastly, since Event I (and Event II) happen with probability (over $\mathcal{T} \sim \mathcal{O}^q$) at most $\delta/(2n)$ w.r.t $v$, the probability of either one of them happening is at most $\delta/n$. It follows that, the probability that $h$ is $(\epsilon/n)$-bad w.r.t $v$ (i.e., $\Pr_{\mathcal{C} \sim \mathcal{D}}[h(\mathcal{C})[v] \neq h^*(\mathcal{C})[v]] < \epsilon/n$) is at most $\delta/n$. That is,

$$\Pr{}_{\mathcal{T} \sim \mathcal{O}^q}[h \text{ is } (\epsilon/n)\text{-bad w.r.t at least one } v \in \mathcal{V}] \leq \delta \quad (14)$$

Thus, the probability (over $\mathcal{T} \sim \mathcal{O}^q$) of $err_{\mathcal{D}}(h) > \epsilon$ is at most $\delta$. This concludes the proof. ∎

**Pseudocode for Algorithm 3.** We present the pseudocode for Algorithm 3 VisRange on learning based on *visiting times of ranges of scores*.

---

**ALGORITHM 3:** `Visiting Ranges` (**VisRange**)

**Input** : A training set $\mathcal{T}$; graph $\mathcal{G}$
**Output:** A system $h$

1   $q \leftarrow |\mathcal{T}|$                                   `// Size of the training set`
2   **for** $v \in \mathcal{V}(\mathcal{G})$ **do**
3      $\lambda_{s_1,s_2} \leftarrow 0$, for $s_1, s_2 = 0, ..., \deg_v + 1, s_1 \leq s_2$  `// The hitting time of range`
4      $a_{s_1,s_2}, b_{s_1,s_2} \leftarrow 0$, for $s_1, s_2 = 0, ..., \deg_v + 1, s_1 \leq s_2$  `// The number of output`
       `state-0's and output state-1's under input scores in range`
       $[s_1, s_2]$
5      **for** $(\mathcal{C}, \hat{\mathcal{C}}) \in \mathcal{T}$ **do**
6         $s \leftarrow \text{score}(\mathcal{C}, v)$
7         $\lambda_{s_1,s_2} \leftarrow \lambda_{s_1,s_2} + 1$ for each range $[s_1, s_2]$ that contains $s$
8         **For** each range $[s_1, s_2]$ that contains $s$: $a_{s_1,s_2} \leftarrow a_{s_1,s_2} + 1$ **if** $\hat{C}[v] == 0$; **else**
         $b_{s_1,s_2} \leftarrow b_{s_1,s_2} + 1$
9      **end**
10     $S \leftarrow \emptyset$
11     **for** $s_1 = 0, ..., \deg_v + 1$ **do**
12       **for** $s_2 = s_1, ..., deg_v + 1$ **do**
13         **if** $\lambda_{s_1,s_2} \geq \frac{\epsilon}{2n} \cdot q$ **then**
14           $\ell'_{s_1,s_2} \leftarrow 0$ if $a_{s_1,s_2} > b_{s_1,s_2}$; Else, $\ell'_{s_1,s_2} \leftarrow 1$  `// Majority voting on`
            `the correct output state of v under input scores in`
            `range` $[s_1, s_2]$
15           $S \leftarrow S \cup \{(s_1, s_2)\}$
16         **end**
17       **end**
18     **end**
19     **if** $\exists (s_1, s_2) \in S$ *s.t.* $\ell'_{s_1,s_2} = 0$ **then**
20       In $h$, set $\tau_v \leftarrow 1 + \max \{s_1 : (s_1, s_2) \in S, \ell'_{s_1,s_2} = 0\}$          `// The learned`
       `threshold`
21     **end**
22     **else**
23       In $h$, set $\tau_v \leftarrow 0$
24     **end**
25 **end**
26 **return** $h$

---

**Detailed proof of Theorem 4.2**

In Theorem 4.2, we prove the sufficient number of training samples needed by Algorithm 3.

**Theorem 4.2** *For any $\epsilon, \delta \in (0, 1)$, and any $\eta_v \leq \bar{\eta} < 1/2, v \in \mathcal{V}$, with a training set of size*

$$q = O\left( \frac{1 - \bar{\eta}}{(1/2 - \bar{\eta})^2} \cdot \frac{1}{\epsilon} \cdot n \cdot \log(\frac{n}{\delta}) \right)$$

*Algorithm 3 (i.e., **VisRange**) learns a hypothesis $h \in \mathcal{H}$ such that with probability at least $1 - \delta$ (over $\mathcal{T} \sim \mathcal{O}^q$), we have $err_{\mathcal{D}}(h) < \epsilon$.*

**Proof.** We show that a training set of size

$$q = 8 \cdot \frac{1 - \bar{\eta}}{(1/2 - \bar{\eta})^2} \cdot \frac{n}{\epsilon} \cdot \ln \left( \frac{2\Delta n}{\delta} \right) \tag{15}$$

is sufficiently large to establish the $(\epsilon, \delta)$-PAC guarantee. Recall that $\Delta < n$ is the maximum degree of the underlying graph. Conceptually, different from the analysis of Algorithm 2 where one cares about the number of times each *score* is visited in a training set $\mathcal{T}$, here in Algorithm 3, we focus on the visiting time of each possible *range of scores*. In particular, for a vertex $v$, each range $R$ of

---

scores with a "high" visiting probability should be visited a large number of times. Consequently, the majority vote over the output states of $v$ in the successors under each score in $R$ would reveal information about the true threshold of $v$, which is captured by Algorithm 3.

**Fix a vertex** $v \in \mathcal{V}$. We consider all possible ranges of scores of $v$, denoted by

$$R_{s_1,s_2} = [s_1, s_2], s_1, s_2 = 0, ..., \deg_v + 1, s_1 \leq s_2$$

We remark that there are $O(\Delta^2)$ such ranges for $v$, where $\Delta$ is the maximum degree of the graph.

Similar to the definition of "importance" for scores, we say that a range $R_{s_1,s_2}$ is $\epsilon$-**important** w.r.t. $v$, if the visiting probability (over $\mathcal{C} \sim \mathcal{D}$) of $R_{s_1,s_2}$ is at least $\epsilon$. Recall that the visiting probability of a range $R_{s_1,s_2}$ w.r.t. $v$ is the probability of sampling a configuration $\mathcal{C} \sim \mathcal{D}$ such that the score of $v$ under $\mathcal{C}$ is in $R_{s_1,s_2}$.

**Fix a range** $R_{s_1,s_2}$ that is $(\epsilon/n)$-important w.r.t $v$; at least one such a range $R_{s_1,s_2}$ exists. Let $X(R_{s_1,s_2}, \mathcal{T})$ be the random variable (over $\mathcal{T} \sim \mathcal{O}^q$) representing *the sum of the visiting times* over the scores in the range $R_{s_1,s_2}$ under training set $\mathcal{T}$. That is,

$$X(R_{s_1,s_2}, \mathcal{T}) = \sum_{s=s_1}^{s_2} X(s, \mathcal{T}) \tag{16}$$

where $X(s, \mathcal{T})$ is the random variable (over $\mathcal{T} \sim \mathcal{O}^q$) that records the visiting time of each score $s$ under training set $\mathcal{T}$. Given that $R_{s_1,s_2}$ is $(\epsilon/n)$-important, the expected value of $X(R_{s_1,s_2}, \mathcal{T})$ satisfies that

$$\mathbb{E}[X(R_{s_1,s_2}, \mathcal{T})] \geq \frac{\epsilon}{n} \cdot q$$

In Algorithm 3, one only cares about the ranges whose visiting time under $\mathcal{T}$ is at least

$$t = \frac{1}{2} \cdot \frac{\epsilon}{n} \cdot q$$

When $q$ is sufficiently large, we want every $(\epsilon/n)$-important range to be visited at least $t$ times so these important ranges will be examined by the learning algorithms. By Chernoff, one can verify that

$$\Pr_{\mathcal{T} \sim \mathcal{O}^q}[X(R_{s_1,s_2}, \mathcal{T}) < t] \leq \exp\left(-\frac{1}{8} \cdot \frac{\epsilon}{n} \cdot q\right) \tag{17}$$

Note that "$X(R_{s_1,s_2}, \mathcal{T}) < t$" happening for the $\epsilon/n$-important range $R_{s_1,s_2}$ is undesirable. We now define the following bad event:

**Event I:** $X(R_{s_1,s_2}, \mathcal{T}) < t$ for at least one $\epsilon/n$-important range $R_{s_1,s_2}$

By Ineq 17, we have

$$\Pr_{\mathcal{T} \sim \mathcal{Q}^q}[X(R_{s_1,s_2}, \mathcal{T}) < t \text{ for at least one } \epsilon/n\text{-important range } R_{s_1,s_2}] \tag{18}$$

$$\leq 2\Delta^2 \cdot \exp\left(-\frac{1}{8} \cdot \frac{\epsilon}{n} \cdot q\right)$$

where the factor $\Delta^2$ comes from the fact that there are $O(\Delta^2)$ such ranges for $v$. Importantly, one can verify that, when the size of the training set $q$ satisfies Eq 15, Event I happens with probability (over $\mathcal{T} \sim \mathcal{O}^q$) at most $\delta/(2n)$.

The second property that Algorithm 3 uses is that, when a range $R_{s_1,s_2}$ is visited a sufficiently large number of times (i.e., at least $t = 1/2 \cdot \epsilon/n \cdot q$ times) over the training set $\mathcal{T}$, if the *majority output state* of $v$ over all the erroneous successors under the scores in $R_{s_1,s_2}$ is 0, then there must exist at least one score $s$ in $R_{s_1,s_2}$ such that the true output state of $v$ in an error-free successor under the score $s$ is also 0.

We now **fix a range** $R_{s_1,s_2}$ that satisfies the following two properties under a training set $\mathcal{T} \sim \mathcal{O}^q$:

**Property I**: Both $s_1, s_2 < \tau_v^*$ or both $s_1, s_2 \geq \tau_v^*$, where $\tau_v^*$ is the threshold of vertex $v$ under the ground-truth system $h^*$

**Property II**: Range $R_{s_1,s_2}$ got visited at least $t = (1/2) \cdot (\epsilon/n) \cdot q$ times.

We remark that such a range $R_{s_1,s_2}$ must exist since the entire range $R_{0,\deg_v+1}$ is visited exactly $q$ times over $\mathcal{T}$. Let $Q(R_{s_1,s_2},\mathcal{T})$ be the visiting times of the range $R_{s_1,s_2}$; $Q(R_{s_1,s_2},T) \geq t$.

Note that, by property $(I)$ stated above, the *true* output state of vertex $v$ under all the scores in $R_{s_1,s_2}$ are the same. Let $\ell_{s_1,s_2}$ denote this true output state of vertex $v$ under the scores in $R_{s_1,s_2}$. That is, when there are no errors, if the input to $v$'s interaction function is any score $s$ in $R_{s_1,s_2}$, then the state of $v$ returned by the ground-truth system $h^*$ is $\ell_{s_1,s_2}$.

Let $Y(\ell_{s_1,s_2},\mathcal{T})$ be the number of training samples $(\mathcal{C},\hat{\mathcal{C}}) \in \mathcal{T}$ such that, the score of $v$ under $\mathcal{C}$ is in range $\mathcal{R}_{s_1,s_2}$, and $\hat{C}[v] = \ell_{s_1,s_2}$ is the succeeding state of $v$. One can verify that

$$\mathbb{E}[Y(\ell_{s_1,s_2},\mathcal{T})] \geq (1-\bar{\eta}) \cdot Q(R_{s_1,s_2},\mathcal{T}) \geq (1-\bar{\eta}) \cdot t$$

where $\bar{\eta} < 1/2$ is the upper bound on the error terms. Ideally, $Y(\ell_{s_1,s_2},\mathcal{T})$ should be strictly larger than $1/2 \cdot Q(R_{s_1,s_2},\mathcal{T})$. That is, the output state $\ell_{s_1,s_2}$ of $v$ appears in strictly more than half of the pairs $(\mathcal{C},\hat{\mathcal{C}}) \in \mathcal{T}$ where $\texttt{score}(\mathcal{C},v) \in R_{s_1,s_2}$. By Chernoff, one can verify that:

$$\Pr{}_{\mathcal{T}\sim\mathcal{O}^q}[Y(\ell_{s_1,s_2},\mathcal{T}) \leq \frac{1}{2} \cdot Q(R_{s_1,s_2},\mathcal{T})] \tag{19}$$

$$\leq \exp\left(-\frac{1}{2} \cdot \left(\frac{(1-\bar{\eta})-1/2}{1-\bar{\eta}}\right)^2 \cdot (1-\bar{\eta}) \cdot \frac{\epsilon q}{2n}\right) \tag{20}$$

Here, the event "$Y(\ell_{s_1,s_2},\mathcal{T}) \leq \frac{1}{2} \cdot Q(R_{s_1,s_2},\mathcal{T})$" happening for label $\ell_{s_1,s_2}$ is undesirable. Now define the following bad event:

**Event II:** $Y(\ell_{s_1,s_2},\mathcal{T}) \leq 1/2 \cdot Q(R_{s_1,s_2},\mathcal{T})$ for at least one range $R_{s_1,s_2}$ with Property I and II.

By Ineq 19, we have:

$$\Pr{}_{\mathcal{T}\sim\mathcal{O}^q}[Y(\ell_{s_1,s_2},\mathcal{T}) \leq 1/2 \cdot Q(R_{s_1,s_2},\mathcal{T}) \text{ for at least one range } R_{s_1,s_2} \text{ with Property I and II}]$$

$$\tag{21}$$

$$\leq 2\Delta^2 \cdot \exp\left(-\frac{1}{2} \cdot \left(\frac{(1-\bar{\eta})-1/2}{1-\bar{\eta}}\right)^2 \cdot (1-\bar{\eta}) \cdot \frac{\epsilon q}{2n}\right)$$

Subsequently, one can verify that for $q$ in Eq 15, Event II happens with probability (over $\mathcal{T}\sim\mathcal{O}^q$) at most $\delta/(2n)$.

Let $h$ be the hypothesis returned by Algorithm 3. We now present the last piece of the proof which shows that $h$ is $\epsilon/n$-good w.r.t $v$ with probability at least $1-\delta$. Recall that $h$ is $\epsilon/n$-good w.r.t $v$ if $\Pr_{\mathcal{C}\sim\mathcal{D}}[h(\mathcal{C})[v] \neq h^*(\mathcal{C})[v]] < \epsilon/n$. The key claim is as follows:

**Claim A.3.** If **both** Event I and Event II do **not** occur under $\mathcal{T}$, then the learned $h$ must be $\epsilon/n$-good w.r.t $v$.

Suppose Claim A.3 is true. We have shown that, under $q$ in Eq 15, Event I (and also Event II) happens with probability (over $\mathcal{T}\sim\mathcal{O}^q$) at most $\delta/(2n)$ w.r.t $v$. Thus, the probability of either one of them happening is at most $\delta/n$. Consequently, $\Pr_{\mathcal{T}\sim\mathcal{O}^q}[h \text{ is } (\epsilon/n)\text{-bad w.r.t at least one } v] \leq \delta$. It follows that, the probability of $err_{\mathcal{D}}(h) > \epsilon$ is at most $\delta$ and the proof of the Theorem is complete.

We now show that Claim A.3 is true. If **both** Event I and Event II do **not** occur under $\mathcal{T}$, then the followings are true:

**Fact 1.** Every $\epsilon/n$-important range is visited at least $t$ times in $\mathcal{T}$.

Let $R$ be the subset consisting of each range $R_{s_1,s_2}$ where $(i)$ both $s_1$ and $s_2$ lie on the same side of $\tau_v^*$, and $(ii)$ $R_{s_1,s_2}$ got visited at least $t$ times (note that $R_{s_1,s_2}$ does **not** need to be $\epsilon/n$-important). Note that the true output state of vertex $v$ under all the scores in $R_{s_1,s_2}$ are the same.

**Fact 2.** For each $R_{s_1,s_2} \in R$, the majority output state of $v$ is the true output state of $v$.

Let $R_0$ and $R_1$ be the subsets of $R$ where the majority output state of $v$ for every range in $R_i$ is $i$, $i = 0, 1$. Let $W$ be the set of ranges (of scores for $v$) that got visited at least $t$ times in $\mathcal{T}$, and the corresponding majority output state of $v$ is 0. We remark that $R_0 \subseteq W$. In Algorithm 3, we effectively choose the range in $W$ with the largest $s_1$ value, denoted by $R_{s_1^*, s_2^*}$ and learn the threshold of $v$ to be $s_1^* + 1$.

Importantly, we observe that such a $R_{s_1^*, s_2^*}$ is **not** in $R_1$ by Fact 2 above. Thus, it holds that

$$R_{s_1^*, s_2^*} \in R_0, \text{ or } s_1^* < \tau_v^* \leq s_2^*$$

Let $\tau_v'$ be an positive integer less than $\tau_v^*$ where $(i)$ the probability (over $\mathcal{C} \sim \mathcal{D}$) of sampling a configuration $\mathcal{C}$ with score of $v$ in range $[\tau_v', \tau_v^* - 1]$ is larger than $\epsilon/n$, and if $\tau_v' \leq \tau_v^* - 2$, then $(ii)$ the probability of sampling a score between $[\tau_v' + 1, \tau_v^* - 1]$ is at most $\epsilon/n$. If no such a $\tau'$ exists, then the algorithm sets $\tau_v = 0$ and one can easily verify that the learned $h$ is $\epsilon/n$-good w.r.t $v$.

Note that the range $R_{\tau_v', \tau_v^* - 1}$ is an $\epsilon/n$-important range w.r.t $v$, and both $\tau_v', \tau_v^* - 1$ lie on the same (left) size of $\tau_v^*$. Thus by Fact 1 and the definition of $R_0$, we have that

$$R_{\tau_v', \tau_v^* - 1} \in R_0$$

Since $R_{s_1^*, s_2^*}$ is the range in $W$ with the largest $s_1$ value, and since $R_0 \subseteq W$ it follows that $s_1^* \geq \tau_v'$, and the learned threshold

$$\tau_v = s_1^* + 1 > \tau_v'$$

As a result, the learned system $h$ can only make a wrong prediction on the output state of $v$ when the score of $v$ under the sample configuration $\mathcal{C} \sim \mathcal{D}$ falls within the range $[s_1^* + 1, \tau_v]$, which happens with probability strictly less than $\epsilon/n$. That is, $h$ is $\epsilon/n$-good w.r.t $v$. This concludes the proof of Claim A.3 and therefore the theorem. ∎

### A.5 ADDITIONAL DETAILS ABOUT THE EXPERIMENTS

**System Specification.** All experiments were conducted on an HPC cluster. Each compute node in a cluster is a 20–40 core Intel(R) Xeon(R) CPU E5-2630 v3 @ 2.40GHz processor with 128–384 GB of memory. To achieve scalability, experiments with different networks were conducted on different compute nodes utilizing the SLURM scheduler. Each job used up to 64GB of memory and 1 CPU core during execution. All scripts were implemented using Python 3.7.

Overall, we generated up to $5,000$ training samples for learning each system under synthetic networks with up to $4000$ vertices and varying density. For other networks, the number of training samples is up to twice the network size.

**Parameter settings of the synthetic networks.**

Table 1: Parameter Settings of the Synthetic Networks

| Parameter | Notation | Parameter Space |
|---|---|---|
| Network Size | $n$ | $\{500, 1000, 2000, 4000, 5000\}$ |
| Average Degree | $d_{avg}$ | $\{5, 10, 20, 30, 40\}$ |
| Noise | $\eta$ | $\{0.05, 0.1, 0.2, 0.3, 0.4\}$ |

**Information about the networks.**

**Synthetic Networks:** We generated $G_{n,p}$ random graphs based on the ER model (Erdös & Renyi, 1959). The values of $n$ are shown in Table 1. By using suitable values of the probability $p$, we also created networks with different average degree ($d_{avg}$) values, as shown in Table 1. Thus, our experiments use both sparse and dense synthetic networks.

**Real-World Networks:** The first real-world network we use, *ca-GrQc*, is a collaboration network within the field of General Relativity and Quantum Cosmology spanning published works from January 1993 to April 2003 (Leskovec et al., 2007). It has $5,242$ vertices and $14,496$ edges. The second real-world network, *USpowerGrid*, has $4,941$ vertices and $6,594$ edges. It contains information on the power grid of the Western States of the USA. Here, vertices represent electrical components

(e.g., transformer, generator) and edges represent power supply lines (Kunegis, 2013). Both of these graphs were obtained from the website (https://net.science) for the `net.science` software tool (Ahmed et al., 2020).

**Execution Times**

Figure 5 shows the execution times of the algorithms over networks of different sizes and densities. We ran each algorithm with 100 training samples to maintain consistency in the study of execution times. In general, V-ERM has the longest execution time, followed by VisRange and VisScore.

This is consistent with the time complexities of the algorithms: $O(n\Delta^2 q)$, and the complexity is reduced to $O(\Delta^2 q)$ upon parallelization, where $n$ is the network size, $\Delta$ is the maximum degree and $q$ is the size of the training set. For sparse networks (where $\Delta$ is a constant), both of these algorithms have a time complexity of $\mathcal{O}(nq)$ (or $O(q)$ by parallelization), as reflected in Figure 5. However, as the network becomes denser, the difference in the second component of the complexity becomes more dominant and this is also reflected in the second panel of Figure 5.

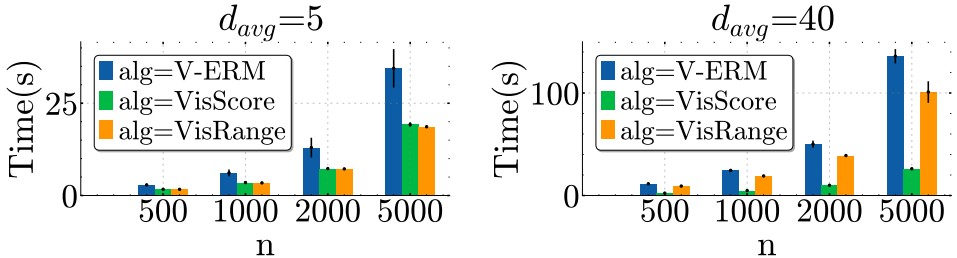

Figure 5: Execution times of the algorithms. The left panel is for sparse networks of various sizes. The right panel is for dense networks. The y-axis scales are different across the two plots. Error bars account for two standard deviation error.

**Additional results**

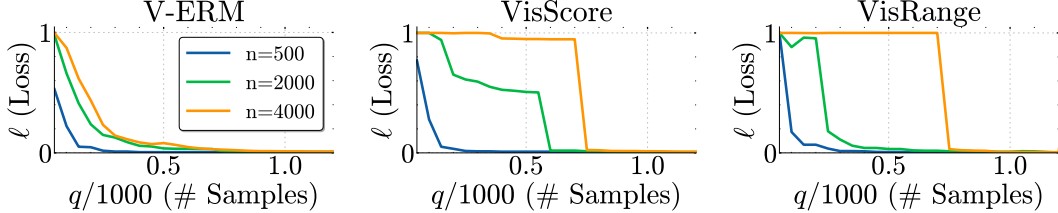

Figure 6: Comparison between the three algorithms on sparse networks in low-noise setting

**Comparison of the algorithms:** Figure 6 compares the learning patterns of all three algorithms. Contrary to the steady nature of V-ERM, both VisScore and VisRange do not learn reasonably until they have a sufficient number of samples, after which they both learn in a phase-transition like manner. It is also observed that for $N = 2,000$, VisRange requires fewer training samples to achieve reasonable performance compared to VisScore.

Figure 7 shows the number of samples required to achieve threshold loss for the two real-world networks. Since the density and network size are fixed, we only present the sensitivity to noise in the figure. Similar to synthetic networks, we find that the required number of samples increases as the threshold loss is decreased. Moreover, scenarios with higher noise need more training samples for achieving the same threshold loss. It is also observed that the required number of samples for *ca-GrQc* is higher than *UspowerGrid* for same parameter setting. Since *ca-GrQc* is larger than *US-powerGrid* both in terms of network size and density, this behavior coincides with that on synthetic networks, described in the main section of the paper.

**V-ERM on synthetic networks:** Figure 8 shows the empirical loss of V-ERM for various synthetic networks. The network size changes along columns and one row of the figure shows networks of same size with increasing density. In general, the learning sensitivity to noise is more prominent

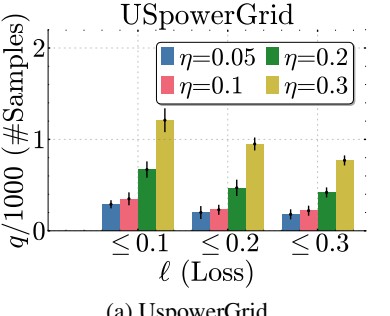

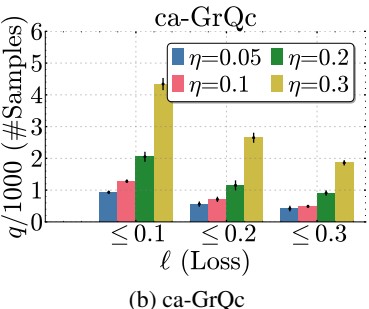

(a) UspowerGrid

(b) ca-GrQc

Figure 7: Training samples needed by V-ERM for achieving specified loss thresholds on real-world networks under various noise settings. Error bars account for two standard deviation errors.

for dense networks than sparse networks. This behavior is present but less prominent for different network sizes with same density.

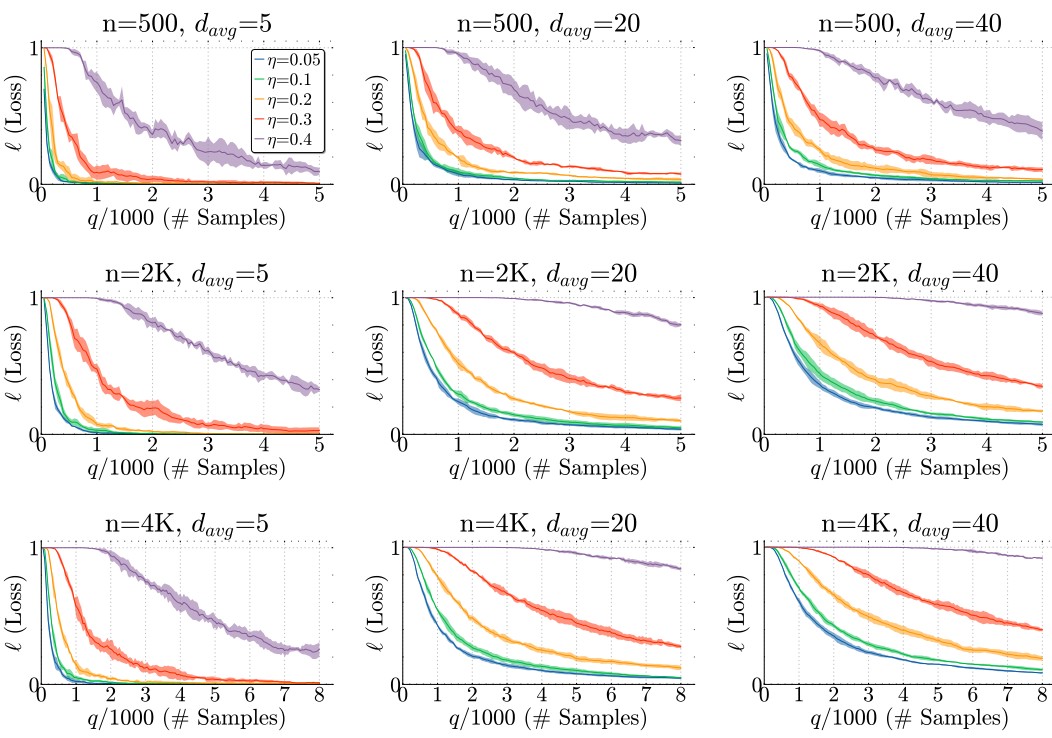

Figure 8: Empirical Evaluation of the learning process of V-ERM (Algorithm 1) under different noise settings for networks of various sizes and densities. Here, $n \in \{500, 2000, 4000\}$ and $d_{avg} \in \{5, 20, 40\}$. The shaded region accounts for 1 standard deviation of the empirical loss.