# OpenReview forum: "Learnability of Discrete Dynamical Systems under High Classification Noise"
_ICLR.cc/2025/Conference — Submitted to ICLR 2025_

### Official Review · Reviewer_xqL6 · 2024-10-26

**Soundness:** 3
**Presentation:** 2
**Contribution:** 3
**Rating:** 5
**Confidence:** 3

**Summary:**

The model being considered is the following: a undirected graph $G=(V.E)$ is given where $V$ are vertices ($n$ in number) and $E$ decides the neighbors of a vertex $v\in V.$ Each vertex $v$ can be in a state $0$ or $1.$  Each node in $v$ has a function $f_v$ associated with it which is appplied sequentially with $t=1,2,\ldots.$

For each iteration $t$, $f_v$ updates the value at the vertex according to the following rule: if the sum of states of neighbors of $v$  at time $t$ is greater than a node dependent threshold $\tau_v$, (not changing with $t$) then the node value at $t+1$ for $v$ will be set to $1$, otherwise, it is set to $0.$

The problem is to learn the threshold values $\tau_v$ correctly from training data for each $v$. The training data assumes that initial state is available and a corrupted version of the transitioned data is available. Non-asymptotic sample complexity bounds are obtained for three algorithms.

The authors move the above dynamic setup to stationary setup wherein the problem is converted to the following. The vertices take values in $\{0,1\}$ with  $C$ representing the configuration  according to a distribution $D$ (which is unknown), a map $h^*$ acts on the the initial state $C$ to provide updated state $C'$ using threshold $\tau_v$ as described erlier. The updated state $C'$ is measured; the measurement adds uncertainty and flips the state (which is either $0$ or $1$) with probability $\eta_v.$ The training set consists of $q$ tuples $(W_i,\hat{W})$ which denote the initial state and the measured transitioned state (which is corrupted). The problem is then to find the optimal  mapping $h$ that minimizes the error with respect to the $h^*$ and obtain the sample complexity of training data needed to provide a needed level of accuracy.

**Strengths:**

The underlying problem being considered is applicable to diverse set of domains. The mathematical approach taken is overall intuitive and seems correct eventhough there are concerns

**Weaknesses:**

The presentation and the main thrust of ideas can be better presented. Some lemmas are not well written or need attention on the mathematics. Please see the questions section of the review to obtain a detailed view of the concerns.

**Questions:**

Detailed comments

$\bullet$ The notation for $h^*$ on line 120 is not in accordance with how its employed in most of the rest of the article.


$\bullet$  PAC system is not described in needed detail.

$\bullet$ Line 170 can be phrased better. Possibly the authors want to empahsize that the learner does not know which transitioned state is wrong. The initial state prior to transitioning is known.

$\bullet$ In Remark 2, authors try and connect the original problem which is dynamic and the "stationary" problem being analyzed. The relationship of the dynamical updates of the original problem and the stationary problem that assumes a distribution of initial states being generated remains unclear and unconvincing; is it possible the dynamic setup cannot be described by a good distribution and thus data collected at diffferent phases of the dynamic evolution will lead to different results.

$\bullet$. In Remark 3, the authors provide rationale on why the measurment noise corruption leading to flipping the bit with probability less than $0.5$ leads no loss of generality. It seems like knowledge whether the probability of flipping the bit is greater than 1/2 or less than 1/2 is needed. The other part of the remark on the difficulty of unravelling the structure of the graph is difficult needs to be better qualified. There is a large literature on estimating topology of agents interacting with each other via quite general relationships. This part of the remark overeaches considerably.

$\bullet$ In line 225, the phrase number of labels is used. There is no explanaation of what labels mean. From later part, it seems like the number of labels is two, either zero or one. It is advisable to remove the use of labels and related development (for example in the Proof of Lemma 3.1) and set it to two.

$\bullet$ On element wise ERM, the authors use  the emperical loss with respect to node $i$ of any hypothesis as

$$\min\sum_{j=1}^q \mathbb{1} \left(h(W_j)[i]\not = \hat{W}_j[i]\right)$$

where the tuples $(W_j,\hat{W}_j)$ are provided to the learner.  The above optimization will lead to a solution say $h^{opt,i}$ for the $i^{th}$ node. The authors seem to make an assumption that $h^{opt,i}=h^{opt,j}.$ Though authors state as much, this assumption is very restrictive.

$\bullet$ Lemma 3.1 is quite confusing. The authors introduce partitions with respect to each node $i$ which are equivalence classes of hypothesis which result in the same empirical loss. Thus, the partition depends on the number $q$ of the available training data and thus are dictated by the probability distributions $D$ and $\eta_v.$ Thus the cardinality of the maximum element of the collection of partitions $t_max$  should be a random variable (dictated by the distribution on the training size, $q$, $D$ and $\eta_v.$) Now the order relation of $q$ involves $t_max$ which is itself dependent on $q$, $D$ and $\eta_v.$ This seems like a circular dependency. Authors needs to indicate clearly what the relationship implies.

$\bullet$ The V-ERM algorithm generates a hypothesis by first setting $\tau^{opt}_i$ the threshold for node $i$ that minimizes the empirical loss for node $i$ and then forming the hypothesis by using $\tau^{opt}_i$ for node $i$ for $i=1,\ldots,n.$ Theorem 3.2 statement does not suffer from the ambiguity of Lemma 3.1 pointed earlier; however, the proof relies on Lemma 3.1. There might be a way to derive Theorem 3.2 directly with the concrete construction of the partition.

$\bullet$ For VisScore and VisRange algorithms, the authors propose to use the training data of configurations and the transitioned measured configurations to evaluate the frequency of a score, of a node, which is number of neighbor states that possess the value $1$ over the $q$ number of training data. For  each of these scores the frequency of the transitioned configuration of the node as measured is also tracked. The statitics of the score and the transitioned value is leveraged to obtain estimates on the threshold to be used for setting the state to one. The analysis tracks scores that have high frequency for a node and the associated frequency of transitioned state being $0$ or $1$. Here standard techniques are employed to arrive at the sample-complexity of etsimates on the probability of scores and the error introduced by the measurement (considered to have a probability less than $1/2$) to diminish with training size $q$ to arrive at the result.

$\bullet$ Simulation results are presented that show $VERM$ showing gradual improvement whereas VisScore and VisRange algorithms show phase transition like behavior. More analysis with respect to the nature of the graph would provide more insights. Authors can clarify how the simulations were carried out; whether the dynamic version was simulated or was distribution of configurations $D$ used in some manner. If the dynamic version was employed then it would be interesting to understand which kinds of distributions arise.

---

### Official Review · Reviewer_Ky3G · 2024-10-28

**Soundness:** 2
**Presentation:** 2
**Contribution:** 2
**Rating:** 3
**Confidence:** 3

**Summary:**

The paper studies the problem of learning discrete dynamical systems under random classification noise. Here, given a graph $G$ with a collection of boolean functions $\{h_v: v\in G\}$, defines the dynamics of the system. Formally, Given $C\in \{0,1\}^{n}$, the next state is generated as $\hat{C}[v]=h_{v}(C)$ for all $v\in G$. The graph to be known and they get samples of the form $C,\hat{C}$ where $C\sim D$ for some distribution $D$. The aim of the task is to predict $\hat{C}$ given $C$ with high probability over $D$.

In this paper, they study the problem when the vector $\hat{C}$ is corrupted by random classification noise. That is, for each vertex $v,\hat{C}[v]=h_{v}(C)$ with probability $1-\eta$ and flipped otherwise. They consider $h_v$ to be the class of 1 dimensional threshold, that is $h_v(C)=1$ iff $\sum_{u\in N(v)}C[u]\geq t_v$ where $t_v$ is a threshold associated to the node $v$ and $N(v)$ is the neighbourhood of $v$ in $G$. They give two algorithms, V-ERM with sample complexity $O_\eta(n^2\log n/\epsilon^2)$ and VisRange with sample complexity $O_{\eta}(n\log n/\epsilon)$, the latter matches the sample complexity of the noiseless case.

They also perform experiments using the algorithm

**Strengths:**

The problem of discrete dynamical systems seems to be an interesting one with practical applications. Robustness to noise in measurements is crucial for safe deployment of such algorithms. The authors take a step towards addressing this problem.

**Weaknesses:**

I might be mistaken but I don't see why the sample complexity of $O(n\log n/\epsilon(1-2\eta)^2)$ doesn't immediately follow as a corollary of prior work on learning with bounded noise in the supervised learning setting. I will sketch an argument here. I am willing to change my score if the authors highlight why the following argument doesn't work and why their theorem is not immediate.

For a VC class of dimension $V$, [MN07] proved that the sample complexity of PAC learning with bounded noise rate $\eta$ scales with $O(\frac{V}{\epsilon(1-2\eta)}\log(1/\delta))$ (see section 1.3.1). Their algorithm is ERM.

Note that since $h_{v}$ is a one dimensional threshold (since G is known), the VC dimension is $O(1)$. Thus, running ERM node-wise should give a sample complexity of $O(\frac{1}{\epsilon(1-2\eta)}\log(1/\delta))$ and guarantees that with probability $1-\delta$, ERM finds a $\hat{h}$ that agrees with $h$ on $1-\epsilon$ fraction of inputs. Setting $\epsilon=\epsilon/n$, $\delta=\delta/n$ and using a union bound gives the desired result. The algorithm is again node-wise ERM.

[MN07] Pascal Massart and Élodie Nédélec; Risk bounds for statistical learning; The Annals of Statistics, vol. 34, no. 5, 2006, pp. 2326–66.

**Questions:**

1) Refer to the Weaknesses section
2) The authors claim that the unknown graph case is hard. I don't see why there isn't a direct reduction to halfspace learning with random classification noise. There are many efficient algorithms for this problem. For the state of the art, see [DKT23]


[DKT23] Ilias Diakonikolas, Christos Tzamos,  and Daniel Kane; A Strongly Polynomial Algorithm for Approximate Forster Transforms and Its Application to Halfspace Learning; STOC 2023

---

### Official Review · Reviewer_ZdWT · 2024-10-30

**Soundness:** 2
**Presentation:** 3
**Contribution:** 2
**Rating:** 5
**Confidence:** 4

**Summary:**

This paper studies the learning of discrete dynamical systems under random classification noise. They consider a dynamical system with a known underlying graph structure with binary values and unknown discrete threshold functions as interaction function. The paper try to answer if the efficient learning is possible under classification noise and establish the required sample complexity for $\epsilon$ prediction error with probability $1-\delta$. They analyze three algorithm: V-ERM, VisScore, and VisRange. V-ERM is element-wise empirical risk minimization algorithm that assigns the threshold value that minimizes the error function over the corrupted training data. The sample complexity scales with $\mathcal{O}(n^2log(n))$ for this algorithm, which is factor $n$ larger than the theoretical upper bound for the noise free case. In addition, they analyze VisRange algorithm which is the extension of the VisScore algorithm. The authors define the notion of visiting time of a score to run this algorithm. For each range score of the vertex, they assign the output based on the majority voting in the training data set and they choose the threshold as the maximum of the left values of the ranges with the majority vote value 0. This algorithm requires $\mathcal{O}(nlog(n))$ number of samples which matches the theoretical upper bound for the noise-free upper bound. Besides the complexity w.r.t. to the dimension of the system, the sample complexities scales with $\mathcal{O}((1-2\eta)^{-2})$ where $\eta$ is the maximum classification noise among the vertices. The author claims that V-ERM performs better in practice that VisRange despite the worse sample complexities. While the algorithm VisRange exhibits phase transition behavior, the loss function for the algorithm V-ERM decreases gradually. Last but not at least, the authors provided numerical experiments with synthetic and real-life data set to support the theoretical results. In their experiments, they tested the evolution of the loss $l$ and the number of required samples with  dependence on the dimension $n$, density of the graph $d_avg$, and the probability of random classification noise $\eta$.

**Strengths:**

- This paper is the first to study dynamical systems under classification noise according to the literature review.
- The problem studied is of significant importance. Although the case with a finite hypothesis class and a single interaction function is analyzed, this paper could be a pioneering work in understanding dynamical systems under classification noise.
- The paper is well-structured and easy to follow in terms of the development of the results. Although there are some minor typos, it is grammatically well-written.
- Remarks provided throughout the paper address potential questions that may arise while reading. I particularly liked the placement of the remarks, as they do not interrupt the flow of the paper.
- The implications of the results and their relation to the existing upper and lower bounds are explained well.
- The proofs are easy to follow and appear mathematically sound in most parts.

**Weaknesses:**

- Many acronyms are used before written in full format. For example, the algorithm names V-ERM and VisRange should be given in full name and what the names stand for must be explained before using the acronyms. Similarly, on line 097, CNF is used without any explanation.
- I think V-ERM is not different from empirical risk minimization, where the Hamming distance between the vectors $h(\mathcal{W}_j)$ and $\hat{\mathcal{W}_j}$ is minimized. V-ERM does not provide any additional insight into the existing literature. Can you clarify how V-ERM differs from the standard empirical risk minimization?
- For line 316, the use of "visiting time" sounds awkward. It implies those are the times the score $s$ was visited, not the number of times the score $s$ was visited. I think it should be renamed as "visiting frequency" rather than "visiting time" because it measures the number of times the score $s$ was visited.
- There are issues with the mathematical representation of a General Learning Model on line 219. You claim it is multi-class learning with $k$ classes for each vertex. However, the vector $\mathcal{W}_j$ takes values in ${0, 1}^n$. It should be ${0, 1, \dots, k-1}^n$. The same error occurs in Appendix A.3 as well. Nevertheless, I believe this does not affect the proofs or mathematical results presented in the paper.
- Despite analyzing the general learning model in section 3.1, the result of Lemma 3.1 does not depend on $k$. I think you must either point out that Lemma 3.1 is for binary values, or you need to include the bound that depends on $k' = (k-2)/(k-1)$, as you provided in the Appendix. Moreover, you can raise $\bar \eta$ up to $(k-1)/(k)$ in the multi-class case. It might be worth mentioning.
- On line 370, you say $n$ is the dominant term, but as $\eta$ approaches $1/2$, the term $\mathcal{O}((1-2\eta)^{-2})$ becomes dominant. I think it is better to say "dependence on the dimension" rather than "dominant term."
- Using the letter $\ell$ for the hypothesis class splits and the loss function value in experiments could be confusing.
- In general, the algorithms seem too simple and appear to have been studied before in various types of classification problems. This problem is a parametric classification problem over a finite number of parameters using a loss function. Therefore, the authors need to explain what kind of new mathematical understanding these algorithms bring. Can you provide a detailed comparison with existing methods in parametric classification?

**Questions:**

Please see the weaknesses above in addition to my questions below.

- Remark 2 is unclear in terms of how this dynamical system could be sampled over a trajectory. I think there could be some issues when the system updates are cyclic and the sampling only captures a single cyclic behavior. Let $\mathcal{D}$ be the trajectory of the system, and suppose the system is sampled every other time period. We can generate a system such that the score for a vertex $i$ is 0 during these time periods, but the threshold function could be arbitrarily large, say, the degree of $i$. Then, you will not be able to learn the threshold function of vertex $i$ despite the minimal training error. In other words, any positive threshold value minimizes the training error, but the test error would be high when we randomly sample from trajectory $\mathcal{D}$. Therefore, I believe there should be additional assumptions on the sampling behavior over a trajectory to avoid such ill cases and cyclic behavior. How can you address this?
- If you are sampling data from a trajectory of the dynamic system, the observed data would be correlated over time. How do you address the correlation over time in your proofs and results? Does it pose an additional challenge? How is it different from samples drawn from independent and identically distributed multiple trajectories of the system?
- If my understanding of Hamming distance ERM is correct, why do you claim ERM cannot be done efficiently unless P = NP on line 064? Do you mean ERM with the error loss function only?
- In Figure 3 (b), the sample complexity does not increase quadratically for the algorithm V-ERM. The sample complexity is $\mathcal{O}(n^2 \log(n))$ for this algorithm. Rather, it looks like a linear increase. How can you express this behavior?
- How do you explain the extension from VisScore to VisRange? What was the theoretical and empirical justification for focusing on this problem?
- How can you generalize these results to hypothesis classes of infinite size, e.g., $|\mathcal{H}| = \infty$? For example, you could have an interaction function with a parameter that can take infinitely many possible values.
- Similarly, how can these results be applicable to a system where vertex $v$ receives faulty observations from its neighbors? In other words, if we did not have random classification noise but had random observation noise from the neighbors due to partial information sharing or information asymmetry, can we use these algorithms for learning?
- How can the result be extended to non-stationary threshold functions (what happens when the threshold changes over time)?

---

### Official Review · Reviewer_mGsE · 2024-11-03

**Soundness:** 3
**Presentation:** 3
**Contribution:** 3
**Rating:** 6
**Confidence:** 3

**Summary:**

This authors study the problem of learning discrete dynamical systems from noisy data. They introduce two algorithms, V-ERM and VisRange, which are noise-tolerant and achieve efficient learning guarantees under PAC-bounds. The paper provides sample complexity bounds and demonstrates that the number of samples needed in the noisy scenario remains in the same order of those in noise-free settings. Experimental results on synthetic and real-world networks further validate the algorithms' effectiveness, revealing performance that favor V-ERM in practical applications and VisRange for theoretical bounds.

**Strengths:**

**Clarity of exposition:** The paper is well-written and systematically introduces the problem setting, contributions, and theoretical derivations. Definitions and assumptions are clearly stated.

**Intuitive and well-discussed results:** The authors provide sound theoretical results with rigorous sample complexity bounds, effectively extending previous works on learning discrete dynamical systems to the noisy setting.

**Theoretical contribution:** Through empirical evaluations, the authors demonstrate the practical relevance of V-ERM in noisy environments and the sharp theoretical guarantees of VisRange, offering both a practical and a theoretically sound solution.

**Weaknesses:**

1)  In Section 2.2, the authors assume that the underlying graph structure is fully known, which simplifies the learning task. Additional discussion on the practical implications would be necessary, especially regarding the scenarios where graph information may only be partially known.

2) The assumption on the noise rate may be too restrictive. The authors could expand more on the implications of $\eta_v = 1/2$.

**Questions:**

How does the guarantees of VisRange vary with increased graph density? It is unclear how the performance scales with denser graphs, as higher connectivity could potentially increase noise propagation. Could the authors give more intuitions on that? Could the authors provide experimental results on how VisRange performs with varying graph densities?

**Minor**: Repeated ``there" in line 1103.

---

### Author Response · Authors · 2024-11-28
**We thank all reviewers for their valuable feedback**

Dear reviewers and ACs,

We acknowledge that significant revisions are needed at this stage. We are currently preparing a revised version of the paper that incorporates all the feedback provided by the reviewers.

Thank you for your valuable feedback! Once the revised version is completed, we will provide detailed responses to the questions raised by each reviewer.

Best,

Authors

---

### Meta-Review · Area_Chair_HPFf · 2024-12-18

**Metareview:**

This work addresses the challenge of learning discrete dynamical systems from noisy data. It introduces efficient noise-tolerant algorithms with provable PAC guarantees and establishes tight sample complexity bounds. The required training samples in the noisy setting match the noise-free upper bound (up to a constant factor) and are only a logarithmic factor higher than the best-known lower bound. Empirical studies on synthetic and real-world networks validate the algorithms' performance.

There reviewers raised several concerns regarding the novelty and technical contribution of the paper. Unfortunately, the authors did not provide any response to these comments.

**Additional Comments On Reviewer Discussion:**

Four reviews were collected for this paper, with three recommending rejection and one recommending acceptance. The AC agrees with the majority vote, supporting a rejection due to the unaddressed critiques raised by the reviewers.

---

### Decision · Program_Chairs · 2025-01-22

Reject